



# The Effects of the COVID-19 Lockdowns on the Composition of the Troposphere as Seen by IAGOS

Hannah Clark[1], Yasmine Bennouna[2,4], Maria Tsivlidou[2], Pawel Wolff[2], Bastien Sauvage[2], Brice Barret[2], Eric Le Flochmoën[2], Romain Blot[2], Damien Boulanger[5], Jean-Marc Cousin[2], Philippe Nédélec[2], Andreas Petzold[3], and Valérie Thouret[2]

[1]IAGOS-AISBL, 98 Rue du Trône, Brussels, Belgium
[2]Laboratoire d'Aérologie (LAERO), CNRS and Université de Toulouse III, Paul Sabatier, Toulouse, France
[3]Forschungszentrum Julich, Institute of Energy and Climate Research 8: Troposphere, Jülich, Germany
[4]Royal Netherlands Meteorological Institute (KNMI), De Bilt, the Netherlands
[5]Observatoire Midi-Pyrénées (OMP-SEDOO), CNRS and Université de Toulouse III, Paul Sabatier, Toulouse, France

**Correspondence:** Hannah Clark (hannah.clark@iagos.org)

**Abstract.** The European Research Infrastructure IAGOS (In-service Aircraft for a Global Observing System) equips commercial aircraft with a system for measuring atmospheric composition. A range of essential climate variables and air quality parameters are measured throughout the flight, from take-off to landing, giving high resolution information in the vertical in the vicinity of international airports, and in the upper-troposphere/lower-stratosphere during the cruise phase of the flight.

Six airlines are currently involved in the programme, achieving a quasi-global coverage under normal circumstances. During the COVID-19 crisis, many airlines were forced to ground their fleets due to a fall in passenger numbers and imposed travel restrictions. Deutsche Lufthansa, a partner in IAGOS since 1994 was able to operate a IAGOS-equipped aircraft during the COVID-19 lockdown, providing regular measurements of ozone and carbon monoxide at Frankfurt airport. The data form a snapshot of an unprecedented time in the 27 year time-series. In May 2020, we see a 39% increase in ozone near the surface

with respect to the 26 year climatology, a magnitude similar to that of the 2003 heatwave. The anomaly in May is driven by an increase in ozone at nighttime which might be linked to the reduction of NO during the COVID-19 lockdowns. The anomaly diminishes with altitude becoming a slightly negative anomaly in the free troposphere. The ozone precursor carbon monoxide shows an 11% reduction in MAM near the surface. There is only a small reduction of CO in the free troposphere due to the impact of long-range transport on the CO from emissions in regions outside Europe. This is confirmed by IASI-SOFRID CO

retrievals which display a clear drop of CO at 800 hPa over Europe in March but otherwise show little change to the abundance of CO in the free troposphere.



## 1 Introduction

The World Health Organization declared the global COVID-19 pandemic in March 2020 (WHO, 2020). The serious threat to
public health led countries to adopt lockdowns and other coordinated restrictive measures aimed at slowing the spread of the
virus. Such measures had an important effect on economic activity and by consequence on the emissions of primary pollutants
from industrial and transport sectors. Much discussed, is the extent to which these lockdowns have had a significant effect on
local air quality and more widely on atmospheric composition and climate (Le Quéré et al., 2020).

Many studies have focused on primary pollutants such as $NO_2$, decreases of which were almost immediately apparent in
satellite imagery from the TROPOspheric Monitoring Instrument (TROPOMI) on the Sentinel-5 Precursor satellite (Veefkind
et al., 2012) over China in January/February, and later over Europe (Bauwens et al., 2020). Emissions of $NO_2$ are strongly
linked to economic activity. Instruments such as TROPOMI have registered weekly cycles of $NO_2$ and drops in $NO_2$ related
to behavioural patterns of work and holiday periods (Beirle et al., 2003; Tan et al., 2009). Thus, the large reductions in $NO_2$
in the tropospheric column during lockdown over the most economically active areas of Europe (in particular the Po val-
ley, Italy) were quickly associated with the drop in industrial output and emissions from transport. Reductions of between
and 38% compared with the same periods in previous years were recorded (Bauwens et al., 2020). However, TROPOMI
is a young instrument, launched in October 2017, and as such, there is not a robust climatology with which to compare
these changes during lockdown, and in particular to control for the influence of different meteorological conditions. Rele-
vant weather conditions might include higher planetary boundary layer heights which drive down the surface concentrations
of pollutants irrespective of any changes in emissions; windy periods, with their impact on the dispersion and deposition of
$NO_2$; and cloudy skies with their impact on satellite retrievals. As TROPOMI is sensitive to clouds, using only cloud-free
columns can lead to a negative sampling bias. In many parts of Europe, skies were unusually clear (van Heerwaarden et al.
(2021); (https://surfobs.climate.copernicus.eu/stateoftheclimate/march2020.php) due to the persistence of anticyclonic con-
ditions and strongly reduced air traffic (Schumann et al., 2021b, a), and a negative bias may have been reinforced during
the lockdown period (e.g. Barré et al. (2021); Schiermeier (2020); https://atmosphere.copernicus.eu/flawed-estimates-effects-
lockdown-measures-air-quality-derived-satellite-observations).

Drops in primary pollutants were also evident from ground-based air-quality networks across Chinese and European cities;
see the review by Gkatzelis et al. (2021) for a comprehensive overview. In Spain's two largest cities, where lockdowns were
extremely strict, the reductions of $NO_2$ concentrations were 62% and 50% (Baldasano, 2020). Lee et al. (2020) calculated an
average reduction of $NO_2$ of 42% across 126 sites in the UK, with a 48% reduction at sites close to the roadside due to the
drop in traffic emissions. Wang et al. (2020b) looked at six different pollutants (PM2.5, PM10, CO, $SO_2$, $NO_2$, and $O_3$) and
found large reductions in $NO_2$ from traffic sources and a smaller reduction in CO from reduced industrial activities in Northern
China. Similarly, Shi and Brasseur (2020) also noted a drop in CO across the monitoring stations in Northern China operated
by the China National Environmental Monitoring Center. Pathakoti et al. (2021), looked at CO from TROPOMI compared with
the climatology from the MOPITT (Measurements of Pollution in the Troposphere) satellite and noted that the CO levels were
lower during the first phase of the lockdown over India but higher during the second phase. This was probably indicative of





the longer lifetime of CO in the atmosphere, and the long- range transport of CO from a variety of global sources. Overall, the reductions in $NO_2$ and CO in near-surface air masses due to COVID-19 lockdown conditions range from 20% to 80% for $NO_2$ and 20% to 50% for CO, for all observations reported globally (Gkatzelis et al., 2021).

The effects on the secondary pollutant ozone, are more complex due to its chemistry. Tropospheric ozone is produced by photochemical oxidation of methane, carbon monoxide, and non-methane volatile organic compounds (NMVOCs) in the presence of nitrogen oxides ($NOx = NO + NO_2$). There is also a contribution from stratosphere-to-troposphere transport (Holton et al., 1995) in certain synoptic situations (Stohl et al., 2005; Gettelman et al., 2011; Akritidis et al., 2018). Near the surface, ozone is lost through dry deposition, titration by NO and reactions with hydrogen oxide radicals (HOx) (Monks, 2005). The fall

in ozone precursors such as CO during lockdown, together with a decrease in available quantities of the NOx catalyst, might have been expected to lead to a fall in ozone. However, Wang et al. (2020b) found that over China, $O_3$ increased, possibly because a lower atmospheric loading of fine particles led to less scavenging of $HO_2$, and greater $O_3$ production as a result. Such effects have been noted over China during the summers of 2005-2016 (Wang et al., 2020a) and so are not unique to the lockdown period. Shi and Brasseur (2020) also found that ozone increased by a factor of two over northern China specifically

noting the wintertime conditions during lockdown. Over southern Europe, ozone was also seen to increase up to 27% in some places, explained by the reduction in NOx and lower titration by NO (Sicard et al., 2020). Ordóñez et al. (2020) cautioned that whilst $NO_2$ fell across the whole European continent, the ozone anomalies were not always of the same sign. Ozone decreased over Spain but increased over much of Northwest Europe where meteorological conditions were favorable for ozone formation, including elevated temperatures, low specific humidity and enhanced solar radiation.

A similar picture is drawn from the collection of data from world-wide near-surface observations as reported by Gkatzelis et al. (2021). The fractional changes for ozone range from a decrease by 20% in Central Asia (4 studies) to an increase of up to 20% for several parts of the world (Africa: 2 studies; South America: 17 studies; West Asia: 17 studies; Southeast Asia: 19 studies). For Europe (134 studies), percentage changes in ozone are on average close to zero with few reported reductions of less than 20% and increases of up to 65%. Although most of the reported data sets include the consideration of meteorological

conditions, this variability highlights the dominant role of the meteorological situation in creating these ozone anomalies at the surface.

Fewer discussions have considered the free troposphere where measurements would be indicative of global or background changes in the levels of pollutants. Steinbrecht et al. (2021) looked at free tropospheric ozone across the northern hemisphere from balloon and sonde measurements from 1-8km in altitude. They found a reduction in free tropospheric ozone of about

7% compared with the 2000-2020 climatological mean which they largely attributed to the reduction in pollution during the COVID-19 lockdowns.

The IAGOS (In-service Aircraft for a Global Observing System) instruments carried on commercial aircraft measure the primary pollutant carbon monoxide and the secondary pollutant ozone, along with water vapour, clouds and meteorological parameters such as temperature and winds (Petzold et al., 2015; Nédélec et al., 2015). Ninety percent of the data are acquired

in the upper troposphere-lower stratosphere (UTLS) when the aircraft attain cruise altitude somewhere between 300 and 180 hPa (9 to 12 km above mean sea level). The remaining 10% of data are collected during landing and take-off over more than





300 airports around the world. During the COVID-19 lockdowns in Europe, there was a large fall in passenger numbers with a consequent impact on the number of IAGOS aircraft flying, and the amount of data collected. However, one of the Lufthansa aircraft was converted to carry cargo, and operated throughout the lockdown period. The aircraft made regular flights from

Frankfurt to Asia carrying important medical supplies. Frankfurt airport has the longest, densest and most homogeneous time-series of all the airports visited by IAGOS. Thus, the climatology calculated there is the most robust (Petetin et al., 2016b) with ozone being measured since 1994 and CO since the end of 2001.

In this article, we present the observed anomalies of both ozone and CO seen over Frankfurt and benefit from the fine 30m vertical resolution throughout the troposphere, to distinguish the surface anomalies from the observations in the free

troposphere. This offers a valuable check on satellite data, and adds unique and valuable vertical information which is not offered by surface sites. We judge the significance of the ozone anomalies against the 26 year climatology (1994-2019) at Frankfurt, putting the observed anomalies in context with other important events such as the heatwave in 2003. To complement the IAGOS data at Frankfurt we use IASI-SOFRID CO retrieval which give an idea of the extent of any regional changes over Europe.

## 2 Data


The research infrastructure IAGOS, is described in detail in (Petzold et al., 2015). Using commercial aircraft as a platform, IAGOS instruments make routine measurements of ozone, and carbon monoxide along with water vapour, cloud particles and meteorological parameters including temperature and winds. A full description of the instruments that measure ozone and CO used here, can be found in (Nédélec et al., 2015). The ozone instrument, a dual-beam ultraviolet absorption monitor has a

response time of 4 s, and an accuracy estimated at about 2 ppbv (Thouret et al., 1998). This 4 second response time corresponds to a vertical distance of about 30 m. In the horizontal, the aircraft covers a distance of about 80km during the first 5km of ascent (Petetin et al., 2018a). Therefore during the ascent and descent phases of the flight, IAGOS provides fine-scale quasi-vertical profiles. Carbon monoxide is measured with an infrared analyser with a time resolution of 30 s (7.5 km at cruise speed of 900 km h-1) and a precision estimated at 5 ppbv (Nédélec et al., 2003)

IAGOS began life in 1994 under the name MOZAIC (Measurement of Ozone and Water Vapour on Airbus in-service Aircraft) (Marenco et al., 1998) and as such IAGOS has provided a long time-series of ozone data over 27 (1994-present) years, and of CO for almost 20 years (2001-present). The homogeneity of the time-series since 1994 has been demonstrated by (Blot et al., 2021), giving confidence that IAGOS data can be used for a robust climatology and for the study of long-term trends. As mentioned above, this gives IAGOS some important advantages over more short-lived data-sets such as those from

satellites, and allows us to put any anomalies into context within the same reference observations.

For the IAGOS measurements, a number of auxiliary, diagnostic fields are delivered with the data as standard level 4 products. These include potential vorticity, geopotential height and boundary layer height which we will use in this article. The boundary layer height which is defined as the boundary layer thickness (zPBL) + orography is calculated by interpolating the European Centre for Medium-Range Weather Forecast's (ECMWF) operational boundary layer heights to the position and





time of the IAGOS aircraft. The ECMWF fields were 1 °horizontal resolution and 3 hour time resolution with 60, 90 or 137 levels in the vertical depending on the time period used (http://www.iagos-data.fr/#L4Place:).

In order to determine the geographic origin and source of the CO measured by IAGOS, a tool known as SOFT-IO (Sauvage et al., 2017a, b) has been developed, that uses FLEXPART (Stohl et al., 2005; Forster et al., 2007) to link the IAGOS measurements with emissions databases via 20-day back trajectories. For the entire IAGOS flight track, SOFT-IO v1.0 (Sauvage et al.,

2017a, 2018) estimates the source region of the CO contribution from 14 different world regions of emissions from the GFAS v1.2. The source regions are as defined by the Global Fire Emissions Database (GFED), although the emissions inventories are GFAS. It can also estimate the contributions from anthropogenic or wildfires. As for the auxiliary diagnostic fields mentioned above, the meteorological data for FLEXPART come from the 1° by 1° ECMWF operational analyses and forecasts with a 6 hour and 3 hour time resolution respectively (Sauvage et al., 2017b).

To set the IAGOS measurements at Frankfurt airport into a regional context, we use CO satellite retrievals from the Infrared Atmospheric Sounding Interferometer (IASI) on the MetOp meteorological platforms (Clerbaux et al., 2009). These retrievals are performed with the SOftware for a Fast Retrieval of IASI Data (SOFRID) described in Barret et al. (2011); De Wachter et al. (2012). This software is based on the RTTOV (Radiative Transfer for TIROS Operational Vertical Sounder) operational radiative transfer code (Saunders et al., 1999; Matricardi et al., 2004) combined with the 1D-Var software (Pavelin et al., 2008).

For CO the SOFRID retrievals provide a maximum of two pieces of information about the vertical profiles from the surface to the lower stratosphere with a maximum sensitivity at about 800 hPa and an estimated error of about 10 % (De Wachter et al., 2012) .

### 2.1   Anomalies of ozone in early 2020

In this first section, we look at the anomalies of ozone which were strongly evident in spring 2020. Figure 1, shows the

seasonally averaged profile of ozone measured at Frankfurt for March-April-May 2020. The data were acquired by one of the IAGOS-equipped Lufthansa passenger aircraft which was based at Frankfurt. It was converted to cargo operations and was kept flying throughout the lockdown period making a total of 84 flights from March-May 2020.

In Fig. 1, the IAGOS observations are marked by the black solid line. The blue solid line represents the seasonal average for the 26 year (1994-2019) reference climatology, and the blue shaded envelope shows the interannual variability of MAM over

this period. Similar averaged profiles were presented in (Petetin et al., 2016b) based on the period 1994-2012. The climatological profiles presented here and in Petetin et al. show the maximum ozone mixing ratios in the free troposphere to be about 60 ppbv increasing from 21 ppbv at the ground and which then increase again in the upper troposphere. In the period MAM 2020, there were notable departures from the climatology. Ozone mixing ratios reached on average 42 ppbv in a layer from the surface up to an altitude of 1000m. Such values are more commonly found during summer heatwaves. In the free troposphere

from 2000-5000m, the abundance of ozone is lower than normal, lying outside the expected interannual range. We consider some possible reasons for these anomalies in the following sections.





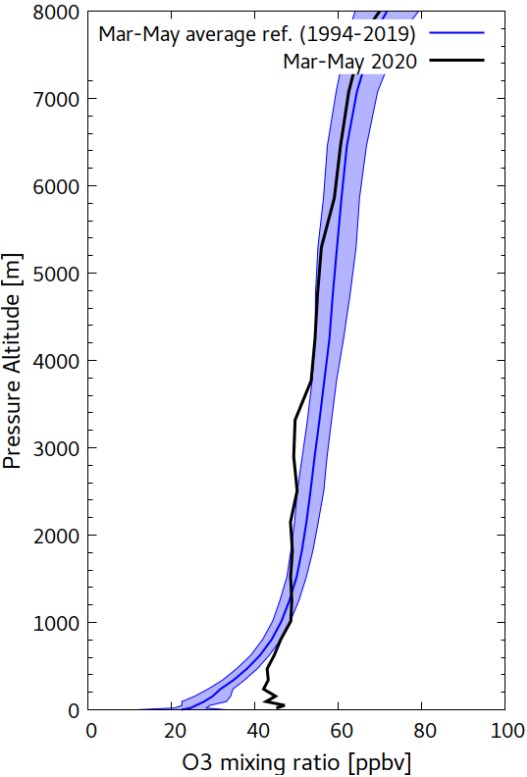

**Figure 1.** IAGOS observations of ozone for March-April- May (MAM) 2020 in black. The blue line is the average profile for MAM calculated over the time-series of the ozone observations (1994-2019). The shaded area represents $\pm 1$ standard deviation of the season MAM over this reference period i.e. the interannual variability of the MAM season.

### 2.1.1 Anomalies of ozone in the surface layer (>950hPa)

The period MAM 2020 corresponded to the period with the most stringent COVID-19 lockdowns across western Europe, but each country had its own date of onset, duration, and different levels of severity. Measures of European mobility (Grange et al.

(2021), based on Google mobility data) reveal that the depths of lockdown were in early April, showing a very slight recovery throughout May. At Frankfurt airport, there was 50% less traffic in March 2020 compared with March 2019, with nearly 80% less traffic in April and May (source, FRAPORT https://www.fraport.com/en/investors/traffic-figures.html, last accessed 18 December 2020). According to Grange et al. (2021) , the restriction measures began in Germany on 22nd March 2020 and had a "Stringency index" defined as "a measure of the strictness of 'lockdown style' policies", that remained relatively high until

the end of May, but that was by no means the strictest in Europe.

In Fig. 2, we present the anomalies in ozone for March and May 2020. We did not have any ozone data in April 2020. It was during the month of May, after the lockdown had been in place for several weeks, when the ozone anomaly in the surface layer (pressure P > 950hPa) was most pronounced (Fig. 2). We require there to be 7 days to make the monthly average otherwise



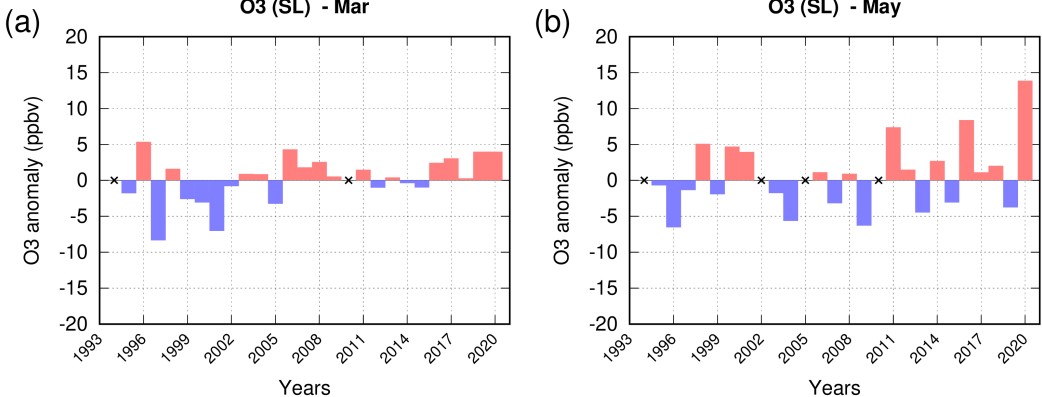

**Figure 2.** Anomalies of ozone for 1994-2020 for all the months of (a) March, (b) May for the surface layer (>950hPa). There were no ozone data in April 2020.

the month is excluded. Excluded months are marked with a cross. In May, ozone was recorded at 13.9ppbv (39%) higher than

the reference average (1994-2019) and was the largest anomaly for the month of May since the time-series began in 1994. The anomaly is apparent in the first 800m of the atmosphere (Fig. 1). A positive anomaly was also observed in March 2020 (15%) with a smaller value compared with May. Positive anomalies in ozone have not been unusual in recent years (see Fig. 3) suggesting that the lockdowns are not the only explanation.

To set the magnitude of these anomalies into context with other periods, the time-series for each month for the surface layer

over Frankfurt, is shown in the top panel of Fig. 3. In addition to the monthly averages, we also show the number of profiles per month as the solid grey bars. During the lockdown period, there were fewer flights than normal and we need to be aware of any sampling bias that this may introduce. The bottom panel of the plot shows a time-series of the monthly anomalies in the surface layer from 1994-2020. There were a number of occasions when the ozone anomalies were comparable to that of May 2020. These were August 2015 and September 2016 when there were short heatwaves, and the other was the well known

heatwave in August 2003 which we discuss here as it was well documented with IAGOS data (Tressol et al., 2008; Ordóñez et al., 2010).

An increase of ozone near the surface can result from increased production of ozone, or reduced sinks of ozone, depending on the conditions and the time of day. Positive anomalies of ozone may be due to an increase in the precursors of ozone, or a prevalence of certain meteorological conditions including increased UV radiation, stagnant airmasses or lower boundary layer

heights which trap the pollutants near the surface. Otherwise, there can be a decrease in the sinks of ozone, such as a decrease in the rate of dry deposition or a decrease in titration by NO due to a reduction in the reservoir of NO. During the 2003 heatwave, IAGOS data showed that there were positive anomalies at Frankfurt in both ozone and the precursor carbon monoxide in the low troposphere, with the ozone anomalies up to 2.5km deep and with the magnitude of the anomalies increasing towards the surface (Tressol et al., 2008). Tressol et al. (2008) found that near the surface, ozone was almost twice the normal amount, and

CO was more than 20% higher. The increased CO was due to the transport of plumes from wildfires over Portugal exacerbated



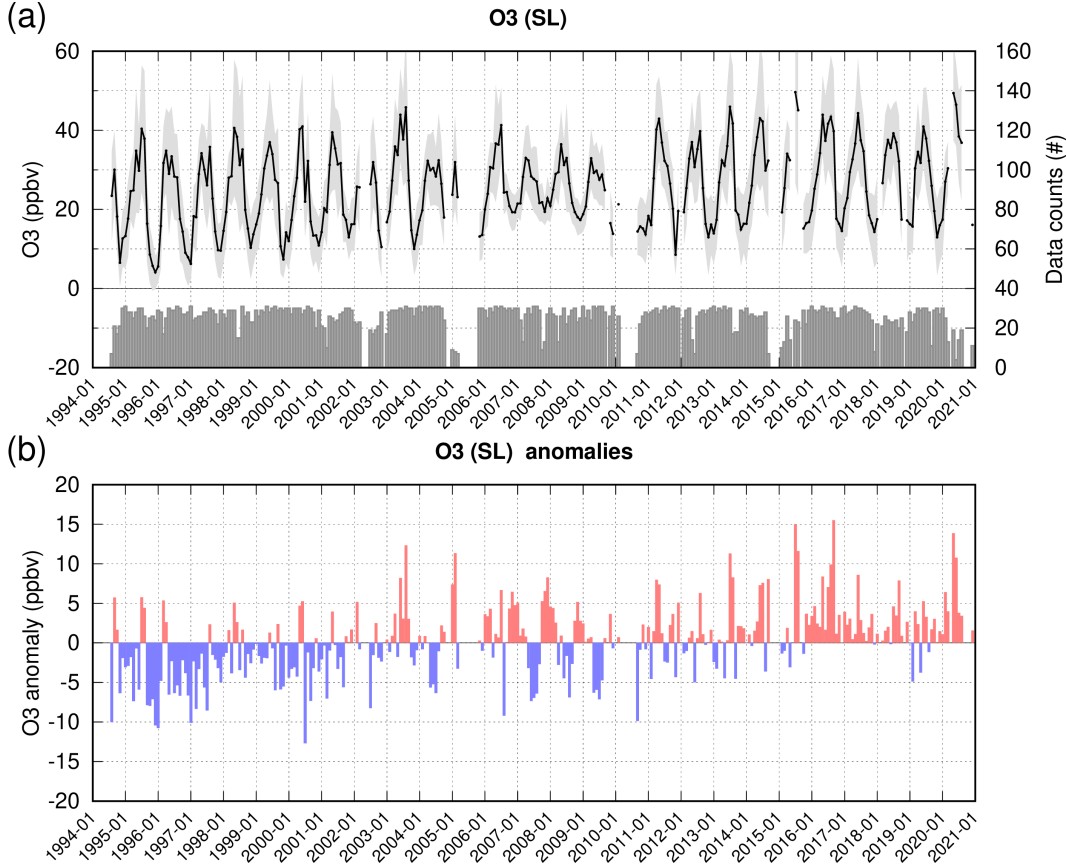

**Figure 3.** Monthly time-series for 1994-2020 of $O_3$ for the surface layer over Frankfurt (a). The grey bars represent the number of daily profiles used to calculate the monthly means shown in black. Grey shading represents the standard deviation of the monthly mean values. (b) The monthly anomalies calculated with respect to the reference average 1994-2019 in the surface layer.

by the dry conditions created by the heatwave. Thus, during the 2003 heatwave, the increased ozone was caused by an increase in precursors and the favorable meteorological conditions.

During lockdown, the chemical environment was quite different. The positive ozone anomaly was accompanied by a drop in the amount of NO as evidenced by the TROPOMI satellite (Bauwens et al., 2020), and there is some evidence from

IAGOS measurements that levels of the precursor carbon monoxide also fell (see section 2.2). The ozone anomaly in the surface layer was most likely due to the combination of increased ozone production due to the exceptionally sunny conditions across a large sector of northern Europe (van Heerwaarden et al. (2021); Ordóñez et al. (2020) , see also $https : //surfobs.climate.copernicus.eu/stateoftheclimate/may2020.php$ ) along with the removal of one of the ozone sinks, particularly the reduction in ozone titration because of the reduction in NO. In addition, the stable meteorological conditions,

lack of wind, and air stagnating over towns, could also have contributed to the accumulation of pollutants and of ozone itself in the boundary layer. Some recent studies have attempted to tease out the contribution of meteorology from the impact of the



changes in the emissions of precursors (Ordóñez et al., 2020; Petetin et al., 2020; Lee et al., 2020). All found that there were important and differing impacts of meteorology, but that the photochemical effects from NOx were dominant.

The magnitude of any anomaly may be significantly influenced by the sampling times within the diurnal cycle. Petetin et al.
(2016a), described the typical diurnal cycle of ozone at Frankfurt airport observed with IAGOS data at different altitudes. They noted that the mixing ratios of ozone are minimum at nighttime due to dry deposition and titration by NO in the shallow nocturnal boundary layer and reach a maximum in the afternoon, due to photochemistry and mixing with ozone-rich layers above the boundary layer. Petetin et al. (2016a) showed the diurnal cycle of ozone at Frankfurt to be maximum between 12:00 and 18:00 UTC in MAM in the layers below 900hPa. The amplitude is maximum at the surface and decreases with altitude,
becoming almost insignificant at altitudes above 900hPa. We consider the anomaly observed in May with respect to the diurnal cycle. More measurements in the afternoon would lead to an oversampling of the maximum and a positive ozone anomaly, and conversely, more measurements at nighttime would be an oversampling of the minimum and a negative anomaly. In Fig. 4, we can see the hourly distribution of the IAGOS profiles for the month of May in 2020 compared with the same month for the reference period 1994-2019. In the climatology, there is a bias towards early morning measurements and in 2020, a bias
towards measurements in early afternoon. This refects the different flight operations carried out during the COVID-19 period. To account for this bias, we calculate the anomaly for May for the daytime (10:00-18:59 UTC) and the nighttime (00:00-09:00/19:00-23:59 UTC) applying this to both the climatology and 2020. These are shown in Fig. 5 (a) and (b) respectively. In Fig. 5 (a) there is still a significant increase of ozone in May during the day (8.3ppbv, 19%), comparable with other anomalies which have occurred 4 times during the last 26 years. This likely reflects the meteorological conditions that were relatively
exceptional and favorable to ozone formation. In Fig. 5 (b) the nighttime increase (12.8ppbv, 41%) is clearly the most significant observed in the time-series highlighting the drop in NO during lockdown and the consequent reduction in ozone titration.

The positive ozone anomaly observed in IAGOS data for the surface layer is in agreement with the other studies cited based on the surface networks and as reviewed by (Gkatzelis et al., 2021). The IAGOS data for the remainder of 2020 (Fig. 3), show smaller positive anomalies which were not significant within the time-series, suggesting that the anomaly in MAM, was
short-lived. We now explore the vertical extent of the ozone anomaly, using the unique perspective that IAGOS offers.



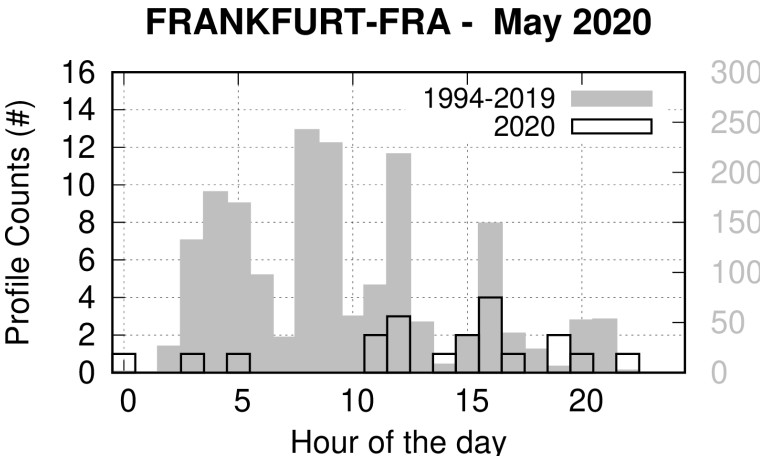

**Figure 4.** Number of profiles by hour of the day (UTC) for May 2020 compared with May over the reference period 1994-2019.

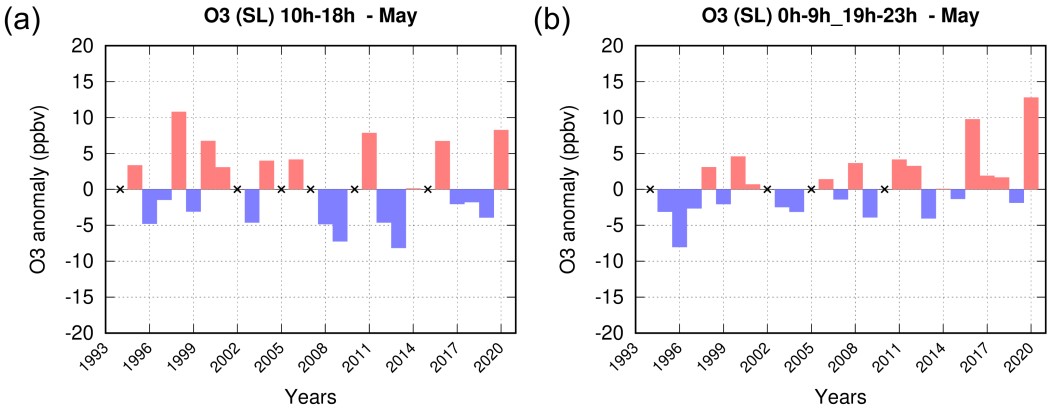

**Figure 5.** Anomalies of ozone for 1994-2020 for all the months of May in the surface layer (>950hPa) for the a) daytime (10:00-18:59 UTC) and b) nighttime (00:00-09:00/19:00-23:59 UTC).



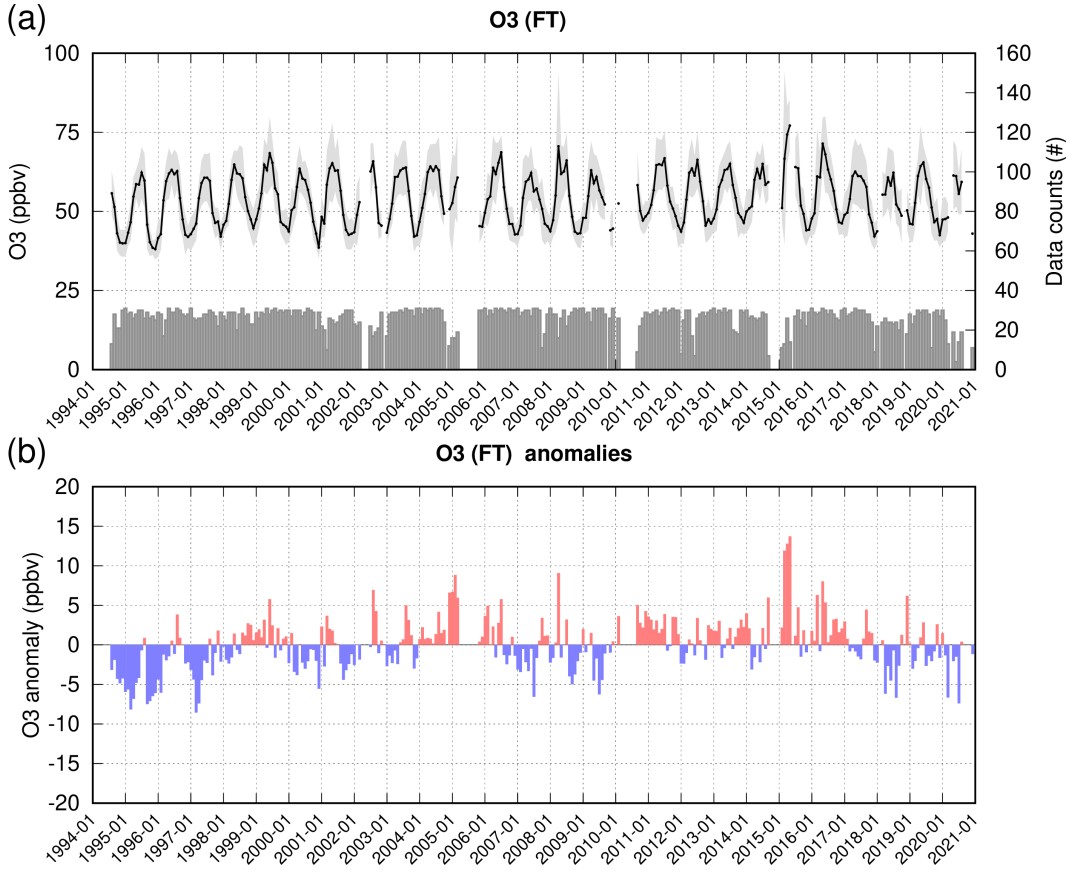

**Figure 6.** Monthly time-series for 1994-2020 of $O_3$ for the free troposphere (850-350hPa) over Frankfurt (a). The grey bars represent the number of daily profiles used to calculate the monthly means shown in black. Grey shading represents the standard deviation of the monthly mean values. (b) The monthly anomalies calculated with respect to the reference average 1994-2019 in the free troposphere.

### 2.1.2 Anomalies of ozone in the free troposphere (850-350hPa)

In contrast to the positive anomaly in the surface layer up to 800m, the anomaly in the free troposphere above 2000m is negative (Fig. 1) lying just on the edge of the range of interannual variability based on the 26 year time-series shown in Fig. 6. The grey bars in the top panel of Fig. 6 represent the number of available daily profiles in each month (where there were more than 7 days available in the month). There was a -6.7 ppbv or -12% drop in ozone in March (Fig. 7). This negative anomaly is the largest for March since 1997 for the IAGOS observations in the free troposphere. It is too early to have resulted from the European lockdowns, and not easy to link with the Asian lockdowns. There was only a 2.0 ppbv (3%) reduction in ozone in the free troposphere over Frankfurt in May 2020 which might be linked to regional European lockdowns (Fig. 7). We had no data for April 2020. The IAGOS data show that ozone levels remained lower than usual for several months after the main lockdown period ended, with an -11% anomaly being observed in July (the largest anomaly recorded for July in the 27 year





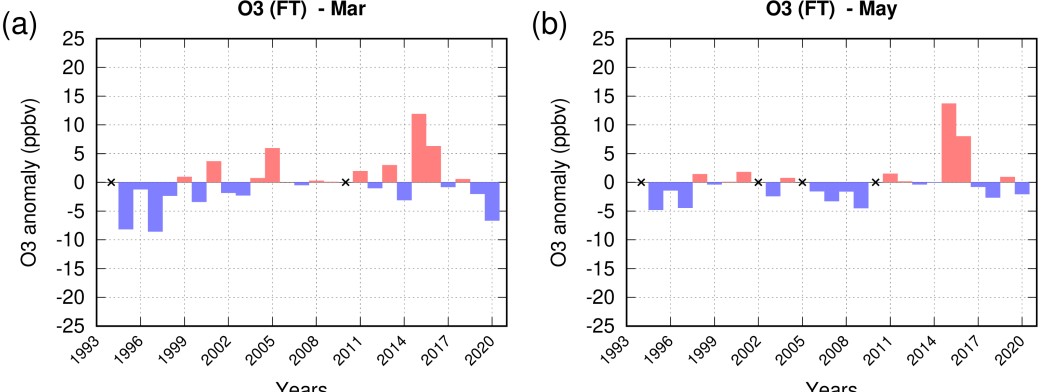

**Figure 7.** Anomalies of ozone for 1994-2020 for all the months of (a) March, (b) May in the free troposphere (850-350hPa). There were no ozone data in April 2020.

time-series; see Fig. A1) as the economic recovery and emissions remained suppressed throughout summer 2020 (Fig. 6). Over the period April-August 2020, a 7% drop was seen by balloons and sondes (Steinbrecht et al., 2021) from 1-8km in altitude. This figure represents a mean value across all sites in the northern hemisphere over the period April-August 2020 compared with the 2000-2020 climatology. The negative anomaly seen at Frankfurt in IAGOS data may illustrate that ozone abundances

fell widely during MAM due to the combined effect of the lockdown measures across Europe, but evidence from the balloon and sonde data (Steinbrecht et al., 2021) suggests that we can expect a degree of geographical variability.





## 2.2 Carbon Monoxide in Spring 2020

As mentioned in the introduction, some studies have demonstrated a fall in the ozone precursors during MAM 2020. The reductions in CO in near-surface air masses due to COVID-19 reported by (Gkatzelis et al., 2021) ranged from 20% to 50% for CO. Due to the long (weeks to months) lifetime of CO in the atmosphere, the causes of these decreases in CO are difficult to attribute. Figure 8, shows the seasonally averaged profile of CO measured at Frankfurt for MAM 2020, with the black, solid line denoting the IAGOS observations. The red solid line represents the seasonal average for the reference period 2016-2019, and the red shaded area shows the interannual variability of MAM over the reference period. Often there are fewer IAGOS observations near the surface than in the free troposphere which leads to the standard deviation being greater near the surface as shown by the wider envelope (shaded red area). The inflexion of the black curve results from a small number of flights which nevertheless fall within the expected range shown by the red shaded area. Despite the length of the time-series being nearly 20 years, we have chosen a shorter segment (2016-2019) for our reference period. This is because there is a negative trend in CO (-1.9%yr-1 and 2.0%yr-1 from Petetin et al. (2016b) for lower troposphere and mid-troposphere springtime respectively) and thus all recent data shows a negative anomaly with respect to the period 2001-2019 (see also Fig. 9). We return to this point later. For the period MAM 2020 we can see that the CO mixing ratios are below the average for recent years (red line) and lie at the lower limit given by the envelope of interannual variability.





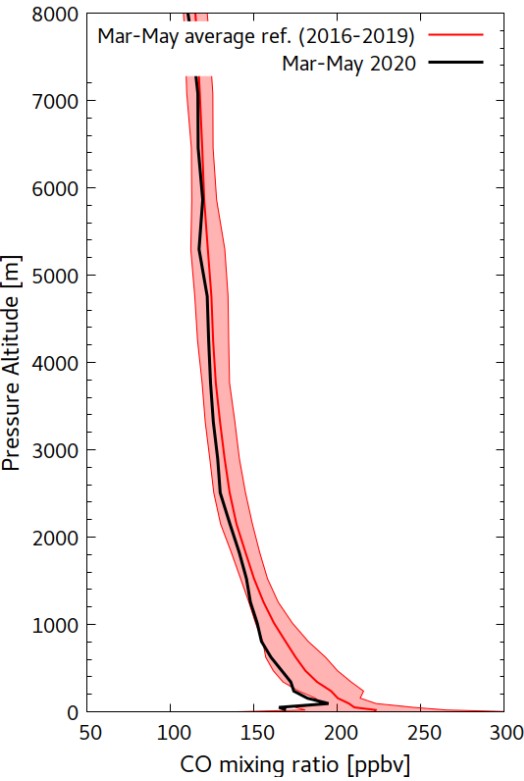

**Figure 8.** IAGOS observations of CO for March-April- May (MAM) 2020 in black. In red, the average profile for MAM calculated over the reference period 2016-2019. The shaded area represents $\pm 1$ standard deviation of the season MAM over this reference period i.e. the interannual variability of the MAM season.

Similar averaged profiles were presented in Petetin et al. (2016b) based on the period (2002-2012) where the average mixing ratio at 2000m for MAM was 150ppbv. For our segment 2016-2019 the average mixing ratio for MAM at 2000m was 140ppbv, indicative of the negative trend of CO. Over the period 2002-2020, there has been a drop in CO mixing ratios observed at
Frankfurt due to a reduction in emissions and the impact of emissions protocols. However, there is a strong interannual variability in the free tropospheric background which reflects the interannual variability of global biomass burning, anthropogenic emissions, and the complex interactions with other species such as OH and $O_3$. The background abundance of CO therefore depends on the selected segment of the time-series.

### 2.2.1 Anomalies of carbon monoxide in the surface layer (>950hPa)

In the surface layer, we see a downward trend in monthly values of CO on the seasonal and annual scale (see Fig. 9) in agreement with Petetin et al., (2016b). Also shown in Fig. 9, is the number of flights per month as the solid grey bars which reveals a reduction in flights due to the reduction in global travel during the first phase of the pandemic. As for ozone, only





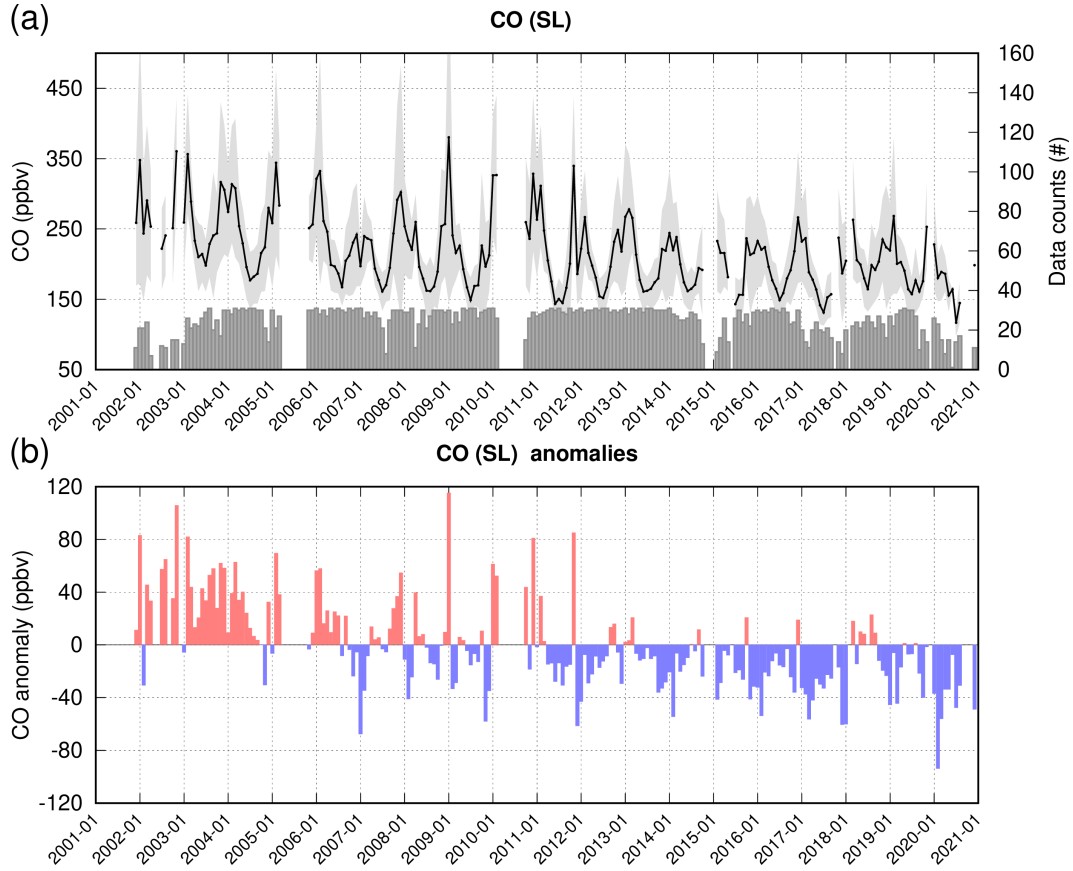

**Figure 9.** Monthly time-series for 2001-2020 of CO for the surface layer over Frankfurt (a). The grey bars represent the number of daily profiles used to calculate the monthly means shown in black. Grey shading represents the standard deviation of the monthly mean values. (b) The monthly anomalies calculated with respect to the reference average 2001-2019 in the surface layer.

months where there are at least 7 days are used to make the monthly average otherwise the month is excluded. The anomalies in Fig. 9 are calculated with respect to the whole length of the CO time-series (2001-2019) and therefore, because of the negative trend, more recent years are all biased low compared with the long term average. The negative trend in springtime over the period 2002-2012 is -1.91%yr-1 (Petetin et al., 2016b). In 2020, we can see a drop in CO mixing ratios with the lowest values in the time-series being recorded in February 2020.

Since recent years are more likely to have a negative bias with respect to the long-term average (2001-2019), in Fig. 10, the anomalies are calculated with respect to the short section of the time-series (2016-2019) as for Fig. 8. Figure 10 shows the anomalies for March, April and May for 2016-2020. In May, towards the end of the lockdown period, the anomaly was (-27.2 ppmv ,-15%) . However, inspection of the time-series (Fig. 9) reveals that the greatest anomaly relative to 2001-2019 was actually apparent in February, before the European lockdown measures began. When compared with the reference time-series (2016-2019), the anomaly in February was ( -61.4 ppmv,-26%, see appendix Fig. B1). February lies outside our main period

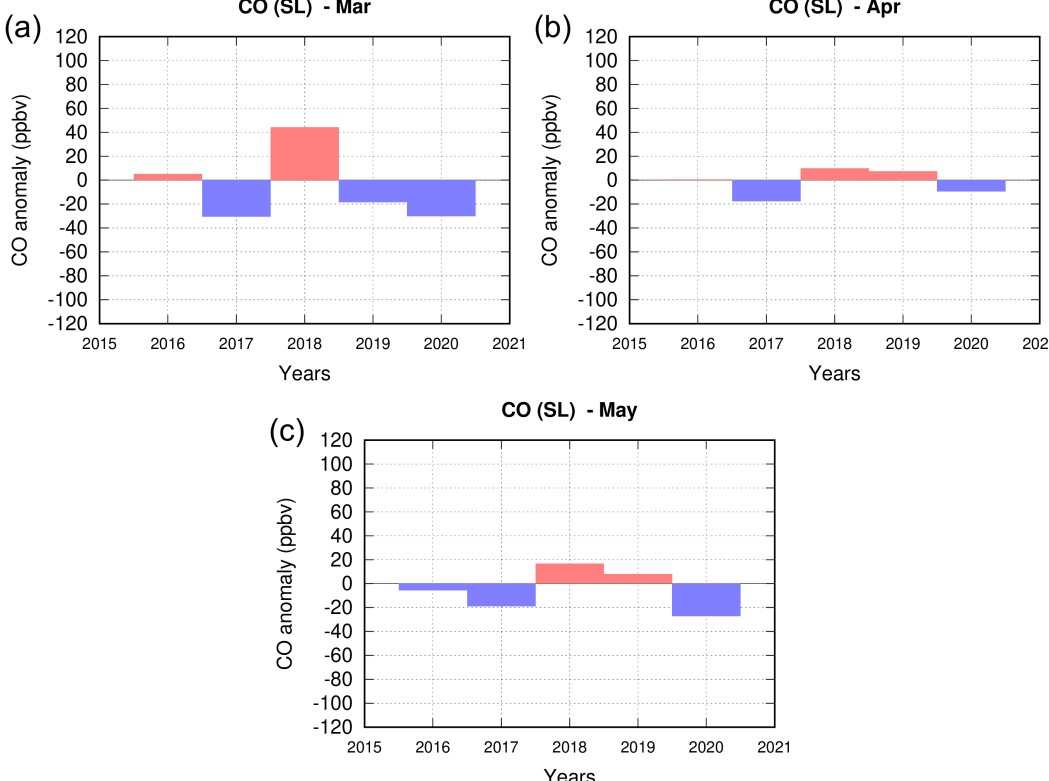

**Figure 10.** Anomalies of CO for 2016-2020 for all the months of (a) March, (b) April, (c) May, in the surface layer (>950hPa).

of interest, but the large anomaly observed then, suggests that the drop in CO at the surface observed in subsequent months,
is not wholly attributed to the drop in emissions linked to lockdown. Indeed, a fall in $NO_2$ levels was also reported by Peuch
(2020) from February onwards. Peuch (2020) suggested that the driver of this could be an increase in boundary layer heights
and consequent dilution of the pollutants near the surface. To investigate if this could be a factor in the low CO seen in IAGOS
data, we examine the boundary layer heights for 2020 compared with the 2016-2019 average.

We calculated the boundary layer height (the boundary layer height (BLH) = the boundary layer thickness (zPBL) + orog-
raphy) by interpolating the ECMWF operational boundary layer heights to the position of the IAGOS aircraft. The ECMWF
fields were 1° horizontal resolution and 3 hour time resolution. This calculated BLH is one of a number of added-value prod-
ucts which are included with IAGOS data as a standard in "level 4". Due to the diurnal variability of the boundary layer height
as the convective boundary layer develops during the day, and the seasonal variability in the time of sunrise and sunset, Fig.
11 is divided into two time slots. The top panel shows the afternoon/evening, defined as 3 hours after sunrise until 2 hours
before sunset. The bottom panel shows the nighttime/morning defined as 2 hours before sunset until 3 hours after sunrise.
Each time-series shows the percentage difference with respect to the monthly mean calculated for 2016-2019 and is shown
for 8 months (January-August) for each year between 2016 and 2019. The depth of the nighttime boundary layer increased by





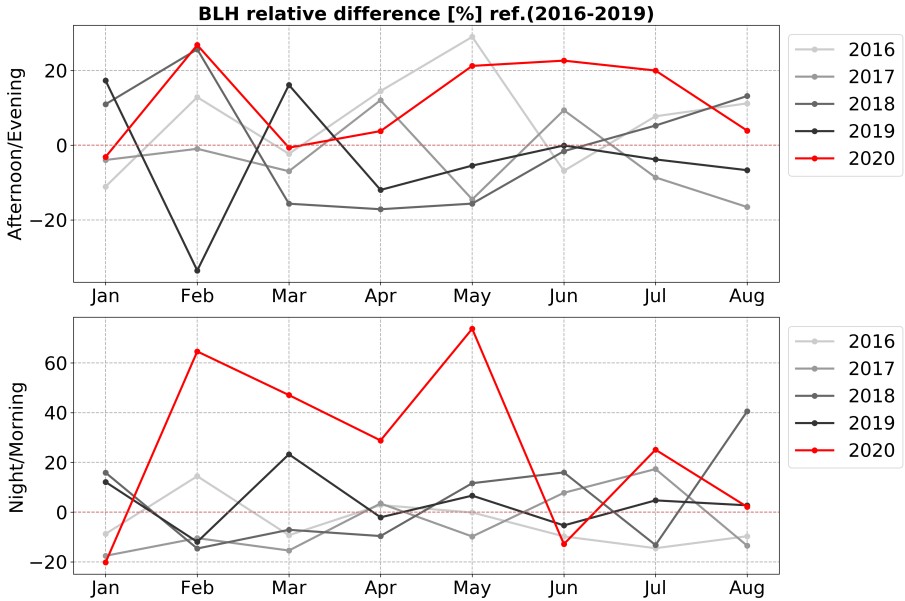

**Figure 11.** Monthly averaged boundary layer heights interpolated in space and time from the 3hourly ECMWF operational fields on $1°$ resolution for afternoon evening (a) and night/morning (b). Afternoon/Evening is defined as 3 hours after sunrise until 2 hours before sunset and Night/morning is defined as 2 hours before sunset until 3 hours after sunrise. The boundary layer heights are expressed as a percentage difference from the mean height calculated over the period 2016-2019.

60% (400m) in February and 70% in May with respect to the monthly mean from 2016-2019 and was greater than any other anomaly observed over the period 2016-2019. The anomaly, covers the MAM period and may partly explain the decreased concentrations of CO in the surface layer.

The observations of CO near the surface from IAGOS are less impacted by the local emissions at airports that might be thought. Petetin et al (2018a), compared IAGOS with monitoring stations from the local Air Quality monitoring network (AQN) and more distant regional surface stations from the Global Atmospheric Watch (GAW) network. They found that the mixing ratios of CO and $O_3$ close to the surface do not appear to be strongly impacted by local emissions related to airport activities and are not significantly different from those mixing ratios measured at surrounding urban background stations. It is therefore unlikely that the reduction in airport activity during COVID-19 was a big contributor to the negative anomaly observed at Frankfurt in the surface layer. In the free troposphere, the local emissions have even less effect and the mixing ratios tend to background concentrations as typically measured by the GAW regional stations.

To examine more closely the source regions of the CO at Frankfurt, we have used the SOFT-IO tool which routinely connects emissions databases to each IAGOS measurement via FLEXPART trajectory calculations (Sauvage et al., 2017b, 2018). However, due to the anthropogenic emissions database (MACC-city) not being updated to take account of the COVID-19 period, we will ignore the indicated contribution from fire or anthropogenic sources given by the link to the emissions databases. We present just the geographic origin of the CO at Frankfurt as determined by the trajectory calculations. The source regions are





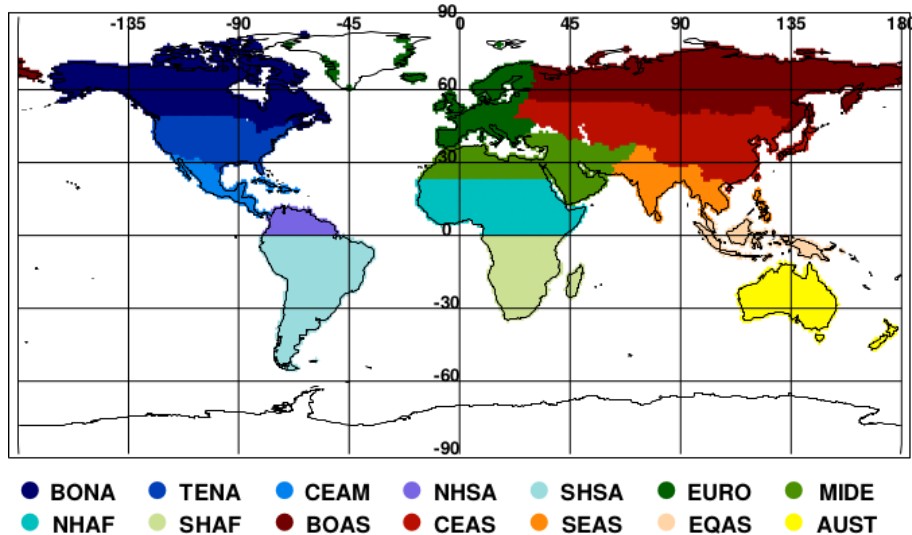

**Figure 12.** Emissions regions used for the SOFT-IO v1.0 model.

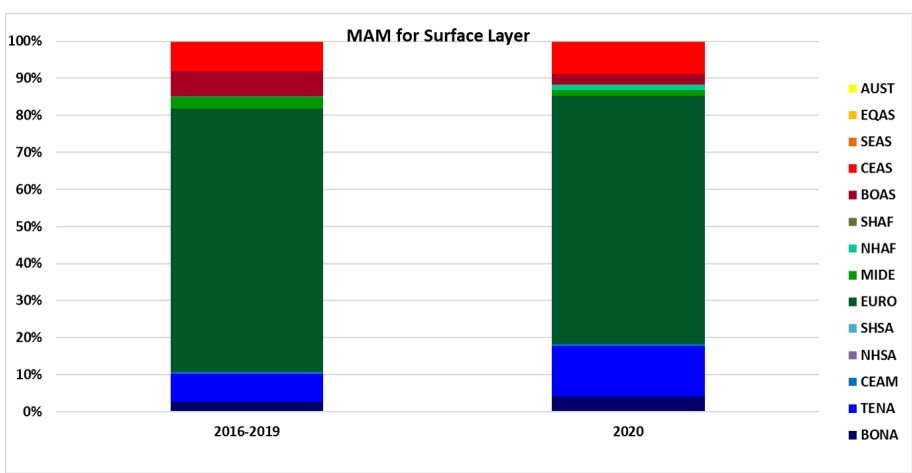

**Figure 13.** Contribution from different source regions to the CO in the surface layer at Frankfurt in MAM 2020 compared with MAM of 2016-2019. The colours correspond to the regions in Fig. 12.

defined as in Fig. 12. We compare the source regions in 2020 with those in our reference period 2016-2019. In agreement with
Petetin et al. (2018b), our analysis shows that the largest contribution to CO measured at Frankfurt is from the European region.
Usually in this period the biomass burning emissions are low, so we deduce that most of the emissions were anthropogenic.
In Fig. 13, we show the source region of the emissions. The majority (70%) of the CO at Frankfurt had a European origin in
both 2020 and in the reference period of 2016-2019. In 2020, there was a greater contribution from sources in North America
(TENA + BONA) and Asia (CEAS) compared with 2016-2019 which reflects the interannual variability of different airmasses





arriving in Europe. This analysis shows that it is primarily local emissions across Europe that are reflected in the CO recorded at Frankfurt in the surface layer, and therefore we can suppose that the lockdown measures played a significant role. In the free troposphere, which we discuss in the following section, we shall see that inter-continental transport has a more important contribution.

### 2.2.2  Anomalies of Carbon Monoxide in the free troposphere (850-350hPa)

In the free troposphere, the anomalies of CO (Fig. 14) were negative in March (-6.9ppbv, -5%) and May (-1.8 ppbv, 1%) but much smaller in magnitude than in the surface layer (see section 2.2.1). The time-series of CO in the free troposphere from 2001-2020 is included for reference in Fig. B2. In April, the anomaly was positive (8.2ppbv, 6%). Since the free troposphere is more representative of the background concentrations due to mixing and transport, it is instructive to relate the IAGOS data over Frankfurt to the larger geographical context. Satellite fields of CO in the tropospheric column are presented for Europe in

Fig. 15. Figure 15 represents the percentage change in the tropospheric CO at 795 hPa with respect to the 2016-2019 average as retrieved from IASI for the months of March, April, and May. In March, the IASI-SOFRID data confirm the negative anomaly in CO present at Frankfurt, which is generalised over large parts of Europe. In April and May, the IASI-SOFRID data showed little anomaly at Frankfurt and a mixed picture over Europe. Thus, the IAGOS and IASI data show some reduction in CO during the lockdown period which is not unexpected given the trend towards decreased CO. It is difficult to link this anomaly

to the lockdown measures due to other factors such as the increased boundary layer height, the long-range global transport of CO, and interannual variability.

Using the same trajectory analysis as for Fig. 13, Fig. 16 shows that in MAM 2020, there was a much lower amount of CO from sources in Europe than in MAM 2016-2019, and an increase in the contribution from North America (TENA) and from Central Asia (CEAS). We suggest that the increase in airmasses from outside Europe negated the effects of the cut in

emissions during the European lockdown resulting in a smaller than anticipated negative anomaly observed by both IAGOS and IASI-SOFRID. This result is similar to that of Field et al. (2020), who noted only a 2% change in background abundances of CO over Eastern China despite the cut in industrial emissions during the Chinese lockdown. In the case of Field et al. (2020) it was the cross-boundary transport from areas with active biomass burning that off-set the drop in anthropogenic emissions.

In summary for this section on CO, we conclude that the drop in surface CO is largely the result of the drop in emissions

during European lockdown, with higher than usual boundary layer heights further diluting the surface concentrations. In the free troposphere, where the negative anomalies were small, the influence of long range transport is more apparent, and offsets the impact of the reduction in CO emissions across Europe.



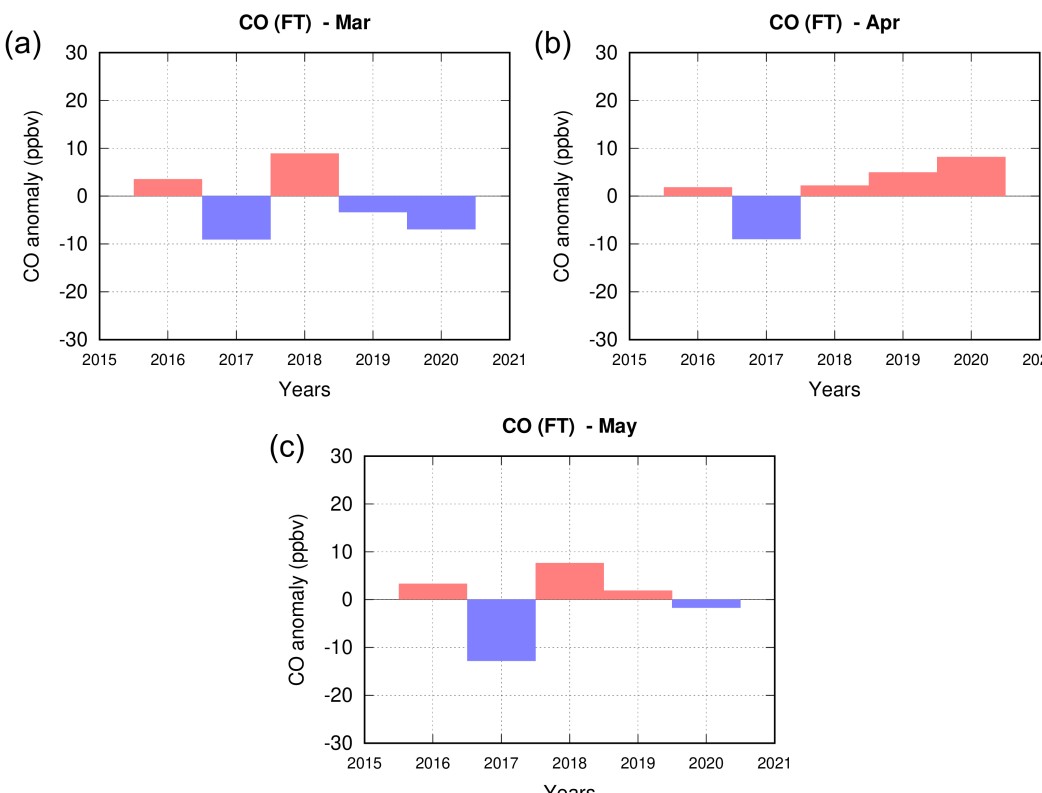

**Figure 14.** Anomalies of CO for 2016-2020 for all the months of (a) March, (b) April, (c) May, in the free troposphere (850-350hPa).

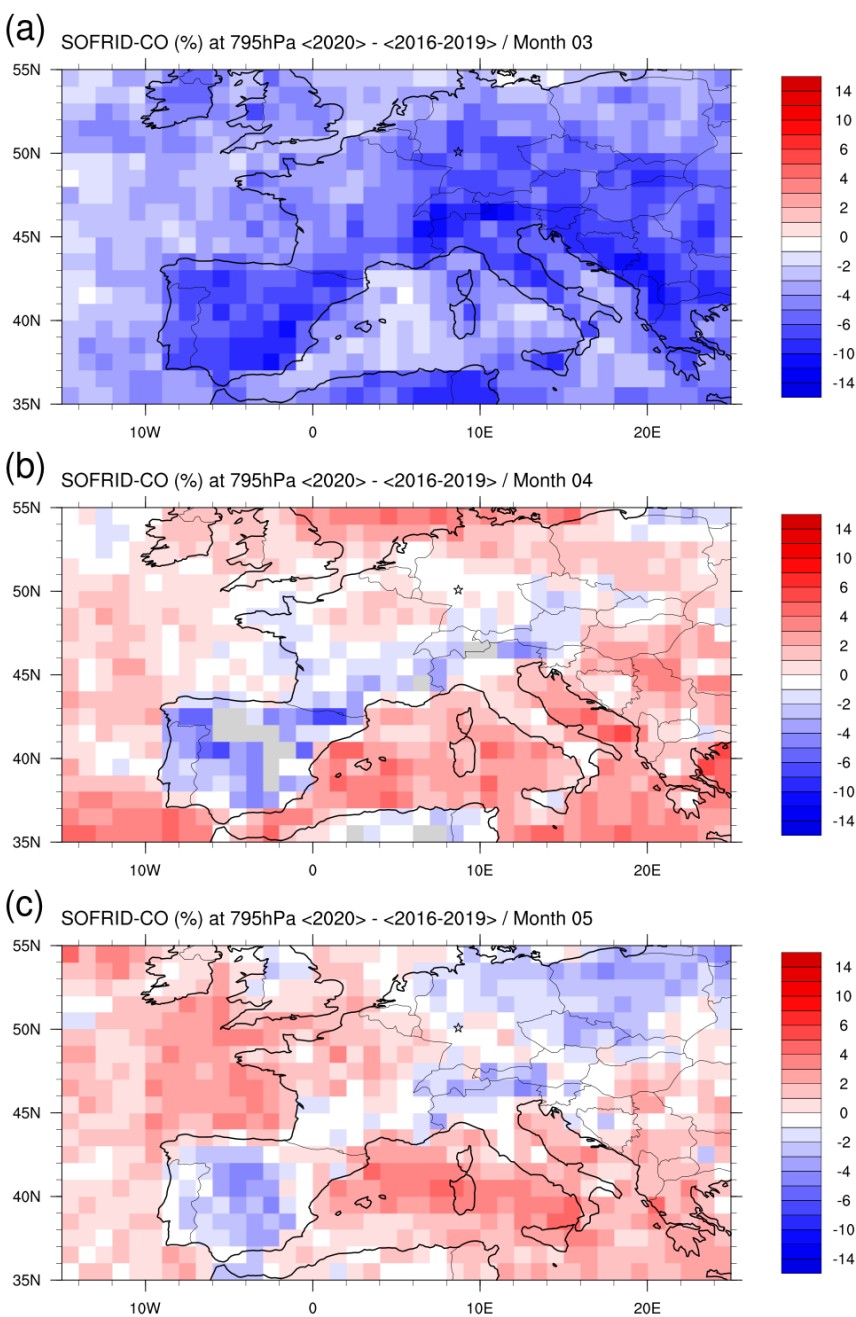

**Figure 15.** Percentage change of IASI-SOFRID Carbon monoxide at 795hPa in (a) March, (b) April, and (c) May, 2020 compared with the reference average period 2016-2019.





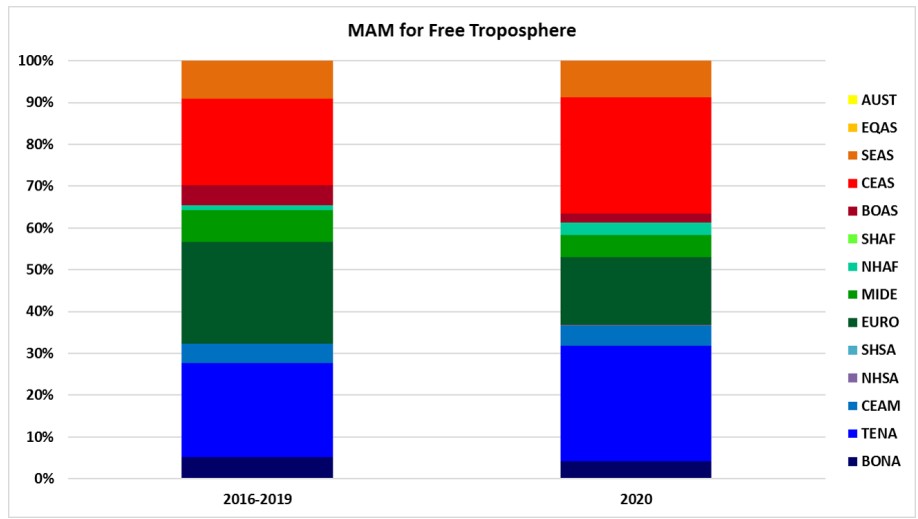

**Figure 16.** Contribution from different source regions to the CO in the free troposphere at Frankfurt in MAM 2020 compared with MAM of 2016-2019. The colours correspond to the regions in Fig. 12



# 3    Conclusions

In this article, we use the IAGOS dataset of in situ observations of ozone and carbon monoxide collected during landing and
take-off at Frankfurt airport. The atmosphere is sampled from the surface to the upper troposphere forming a quasi vertical
profile. The data form part of a time-series which extends back for 27 years for ozone and 20 years for CO. During the
springtime (MAM) of 2020, we noted a 27% increase in ozone in the surface layer (> 950hPa,  600m). The month of May saw
the largest anomaly since the time-series began 1994. The magnitude of this anomaly is comparable to the European heatwave
in August 2003. As this anomaly occurred during the first lockdown period of the COVID-19 outbreak, we thus considered if
the anomaly was related to the changes in emissions that resulted from the decreased traffic and industrial activity. The daytime
increase was significant (19%) but the increase at nighttime (41%) was double and has not been seen before in the 26 year
climatology. Despite the fall in the abundance of NOx over Europe (as seen by satellite data), there were still enough available
precursors to produce ozone under the meteorological conditions that were very favorable, especially enhanced solar radiation,
that prevailed at the time (van Heerwaarden et al., 2021). Our larger increase at nighttime suggests that less ozone was lost
through titration with NO due to the reduction in the NO reservoir during the lockdown period, signifying a reduction in the
"sink" of ozone.

In the free troposphere, ozone abundances fell slightly. For the period MAM 2020 ozone was 5% lower than the mean
1994-2019 based on the same months. The IAGOS time-series shows that these free tropospheric abundances of ozone were
the lowest since 1997 which is probably more reflective of the widespread reduction of emissions over Europe and beyond,
with less impact of local meteorology and chemistry. The results from IAGOS are also consistent with those from the sonde
and balloons reported by Steinbrecht et al. (2021) who noted a 7% drop in tropospheric ozone compared with the 2000-2020
climatological mean. They also attributed this to the reduction in pollution during the COVID-19 lockdowns.

A reduction of CO was seen at Frankfurt, with an 11% reduction found in the surface layer but no anomaly in the free
troposphere as averaged over MAM. Our trajectory analysis shows that the CO is largely of European origin and therefore we
suggest that the anomalies in the surface layer are directly linked to the drop in emissions across Europe due to the lockdown
measures. The increase in boundary layer height explains the onset of the anomaly before the onset of lockdown and probably
contributed to a further dilution of the CO at the surface.

In the free troposphere there is a small reduction of CO in March (-6.9 ppbv, -5%) and May (2 ppbv, 1%) which is smaller
than at the surface over Frankfurt. IASI-SOFRID fields of CO show a clear decrease of CO during March over all of Europe. In
April and May small positive and negative anomalies are detected by IASI-SOFRID over northern Europe and mostly negative
anomalies over the Iberian Peninsula. In the free troposphere, there is an important role of transport of CO from distant biomass
and anthropogenic sources. In particular in MAM 2020, there was a greater contribution from CO originating in North America
and Asia which off-set some of the reduction in regional European emissions, with the result that the lockdown measures did
not have a big impact on CO in the free tropopshere.

The lockdowns provided a unique experiment to assess the impact of a reduction of economic activities on atmospheric
composition and climate. The IAGOS data complement other in-situ data from the ozonesonde network, with the added value





of having ozone precursors measured simultaneously. This study highlights the importance of long and continuous time-series in setting this brief period in context since there are many competing factors and it is difficult to attribute a single cause. We look forward to future model sensitivity studies to separate these factors and to provide a more realistic magnitude of the impact

of lockdown on the environment.

*Data availability.* The IAGOS data are available through the IAGOS data portal at https://doi.org/10.25326/20 (Boulanger, 2020). The IAGOS time series data set used for this analysis is referenced at https://doi.org/10.25326/06 (Boulanger et al., 2018). The SOFRID-O3 and CO data are freely available on the IASI-SOFRID website (http://thredds.sedoo.fr/iasi-sofrid-o3-co/, last access: 31st May 2021; SEDOO, 2014).

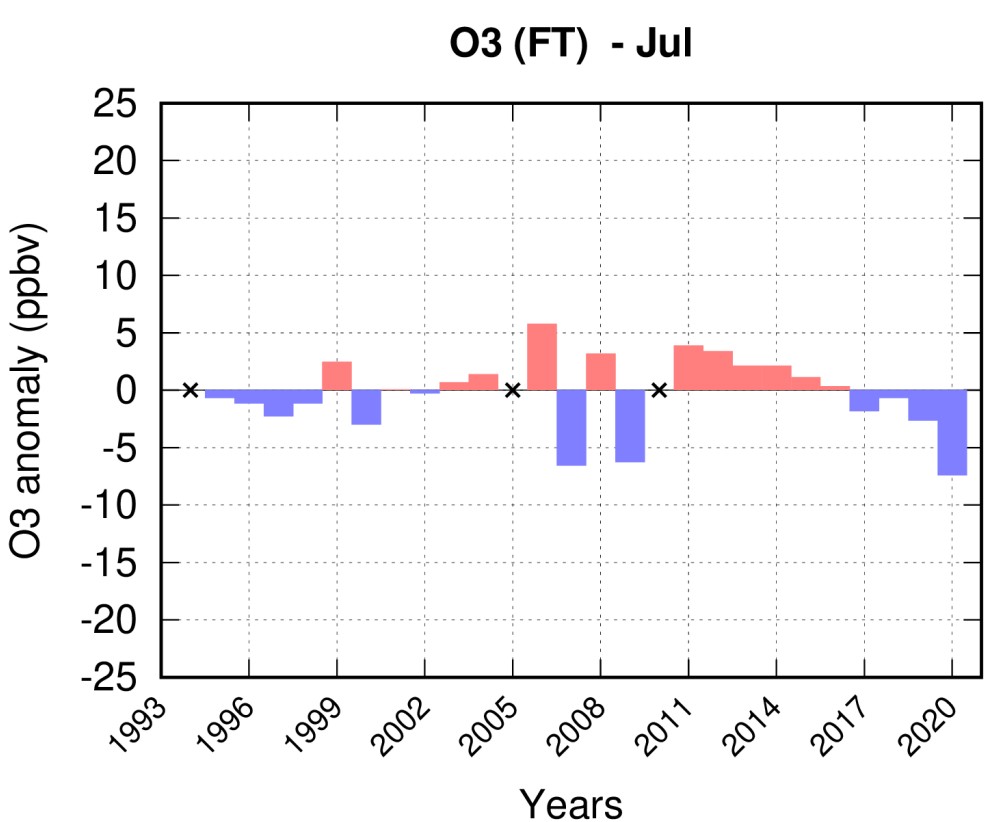

**Figure A1.** Ozone anomalies in the free troposphere (830-350hPa) for the all the months of July since 1994.

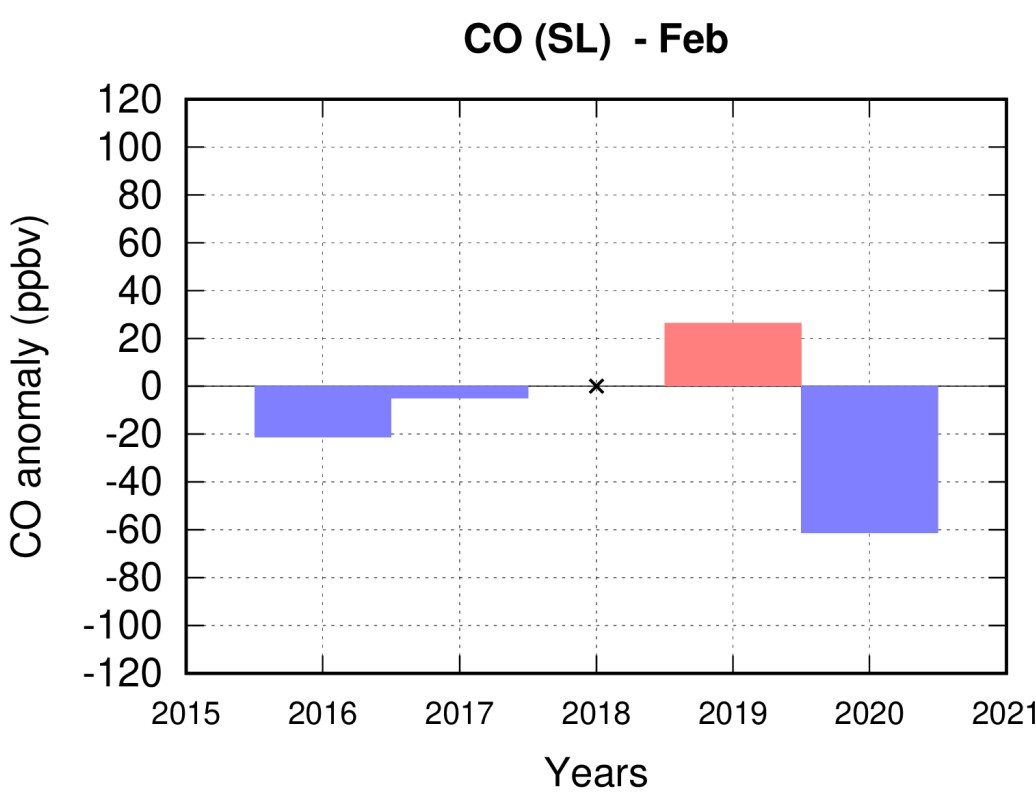

**Figure B1.** Anomalies of CO from 2016-2019 for all the months of February since 2016 in the surface layer (>950hPa).



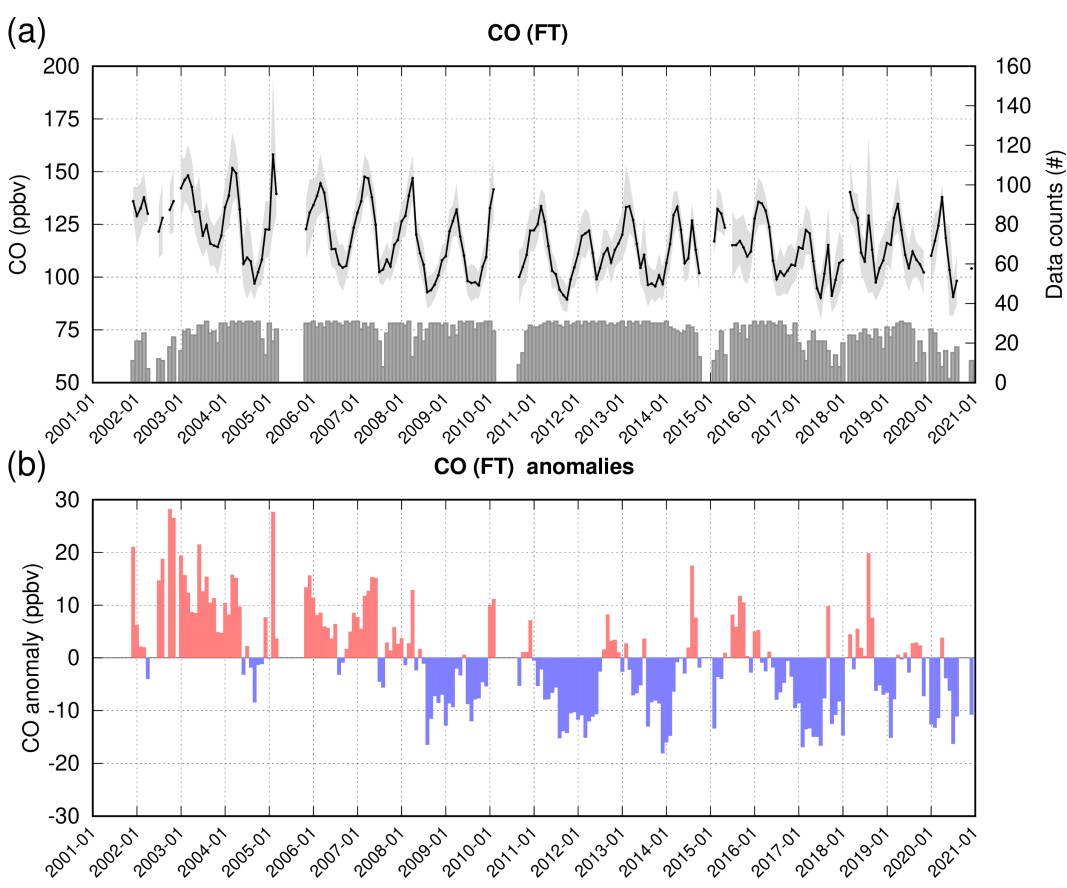

**Figure B2.** Monthly time-series for 2001-2020 of CO for the free tropopshere over Frankfurt (a). The grey bars represent the number of daily profiles used to calculate the monthly means shown in black. Grey shading represents the standard deviation of the monthly mean values. (b) The monthly anomalies calculated with respect to the reference average 2001-2019 in the free troposphere (830-350hPa).



*Author contributions.* HC, YB, and VT designed the study and prepared the paper with contributions from all co-authors. YB developed the code in the framework of the CAMS-84 project. MT, PW, BS are responsible for the level 4 value added products. PhN, RB and JMC produced, validated and calibrated the IAGOS data. DB develops and maintains the IAGOS database. EF and BB performed the IASI-SOFRID retrievals. HC, YB, BS, PhN, VT and AP reviewed and edited.

*Competing interests.* The authors declare that they have no conflict of interest.

*Acknowledgements.* IAGOS has been funded by the European Union projects IAGOS–DS and IAGOS–ERI. The IAGOS database is supported in France by AERIS (https://www.aeris-data.fr). We acknowledge the strong support of the European Commission, Airbus and the airlines (Deutsche Lufthansa, Air France, Cathay Pacific, Iberia, China Airlines and Hawaiian Airlines) that carry the IAGOS equipment. In particular we would like to thank Deutsche Lufthansa for operating D-AIKO in a cargo configuration during the COVID-19 period. IASI is a joint mission of EUMETSAT and the Centre National d'Etudes Spatiales (CNES, France). The authors acknowledge the CNES for financial
support for the IASI activities and the Bonus Stratégique programme at Université Paul Sabatier Toulouse III who fund Maria Tsivlidou.



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
