# Peer review of "The Effects of the COVID-19 Lockdowns on the Composition of the Troposphere as Seen by IAGOS at Frankfurt."

_Atmospheric Chemistry and Physics, 2021_

## Referee Comment (RC2)

ACP Review
The Effects of the COVID-19 Lockdowns on the Composition of the Troposphere as Seen by IAGOS
Clark *et al*
MS No.: acp-2021-479

**Summary**

The authors use ozone, CO and meteorological measurements from the IAGOS dataset for Frankfurt airport to study the impact of COVID lockdowns on air quality. They compare measurements during spring 2020 (European COVID lockdown) to the previous 27 years of data (20 years for CO). They find surface level increases of ozone during the lockdown period driven by increases in nighttime ozone. The anomaly in ozone turns to a slight negative at higher altitudes.
The authors also find a reduction on CO at the surface and little to no reduction at higher altitudes which they attribute to incoming emissions from outside Europe. This hypothesis is strengthened by IASI CO retrievals of the same period.

**General Comments**

This is a decent study that fits well with other COVID air quality related publications. It is useful to have the view of the free troposphere as most publications focus on surface level impacts. The authors give a good overview of the current literature and highlight what extra information this study can give. I feel the following comments need to be addressed before publication.

As the ozone data for April 2020 is missing, is a comparison to the MAM climatology valid? Do you remove April from the previous years to take the climatology and if not, does it have an impact on the result?

I find the difference in time series and calculated climatology between ozone and CO leads to an incoherent picture. I would recommend using the same baseline years for both ozone and CO.

A measure of statistical significance is needed on the presented results. Without this, the arguments for why we are seeing the anomalies aren't as strong as they could be.

I am unsure if using the standard deviation as the definition for interannual variability in this case is the best way to present the results. Would the range or a percentile (e.g. 95[th]) be a more suitable parameter to use? If a month is one standard deviation from the mean there is still ~30% chance it would be completely expected. Showing if the lockdown period fell outside the 95[th] percentile for example would really highlight if it was an unusual year.

Some of the discussion points are mixed in with the results (e.g. the discussion about previous ozone events during heatwaves). This is ok to put the results into context but it leaves the final section of the paper more like a summary rather than a conclusion. I recommend moving some of these sections into the conclusion.

Section 2.2.2 discusses the drop in CO in the free troposphere and trying to relate that to the lockdowns but as far as I can see from figure 8 they were mostly well within the expected interannual variability. This doesn't appear to be mentioned and the results are presented in a way that suggests that the changes are abnormal. Clearly stating the anomalies are within the expected range is needed here.

**Specific & Technical Comments**

There are a number of occasions where there are double negatives (e.g. -XX% drop). Either say XX% drop or -XX% change (or something similar).

Throughout: There are a number inconsistencies with spaces between numbers and units which need to be fixed

Line 23: The Le Quere paper only focuses on CO2, I would either add examples of AQ papers by for the atmospheric composition statement, or rewrite this sentence.

Line 79 & 232: should be 'balloon-borne ozonesondes'

Line 80:  I think this should be 'reduction in surface emissions' not 'pollution'

Figure 1 & 8: I suggest changing the y axis to hPa and mark where your free troposphere definitions begin/end. This way the reader can more easily see which bit of the profile you are discussing.

Lines 156 & 157: Specify air traffic

Lines 156 – 159: is the 50% reduction in (air) traffic related to the the introduction of restriction measures (22$^{nd}$ March) or were there measures in place before this date that was reducing air traffic?

Line 212 & figure 5: 00:00-09:00/19:00-23:59 is confusing, I would either replace the slash with an ampersand or say '7 pm to 9 am'

Figure 4: What do the numbers on the secondary y-axis denote?

Line 245: This should be 'inflection' not 'inflexion'

Line 262: I would say 'As with ozone…' not 'As for ozone…'

Line 291: airports *than* might be though

Line 302/303:  What do you mean you're ignoring fire or anthropogenic sources of CO? That doesn't leave much. Or do you mean you are not distinguishing between them and only looking at source regions? This needs to be clearer.

Line 305: there is reference to fig 13 before saying you're talking about figure 13.

---

## Author Comment (AC1)

Atmos. Chem. Phys. Discuss., referee comment RC1
https://doi.org/10.5194/acp-2021-479-RC1, 2021

[Figure]

**Reponse to Reviewer Comments on acp-2021-479**
Anonymous Referee #1

Referee comment on "The Effects of the COVID-19 Lockdowns on the Composition of the Troposphere as Seen by IAGOS" by Hannah Clark et al., Atmos. Chem. Phys. Discuss., https://doi.org/10.5194/acp-2021-479-RC1, 2021

Review of acp-2021-479 "The Effects of the COVID-19 Lockdowns on the Composition of the Troposphere as Seen by IAGOS"

Hannah Clark, Yasmine Bennouna, Maria Tsivlidou, Pawel Wolff, Bastien Sauvage, Brice Barret, Eric Le Flochmoën, Romain Blot, Damien Boulanger, Jean-Marc Cousin, Philippe Nédélec, Andreas Petzold, and Valérie Thouret

**Summary and General Comments:**

The authors report IAGOS measurements from Frankfurt airport of ozone (data since 1994) and CO (data since 2001) during the MAM 2020 COVID lockdowns. In addition, IASI SOFRID CO satellite data, and ECMWF (boundary layer heights, FLEXPART trajectories) are used in supporting analyses. In general, the authors show increases in March and May 2020 surface layer ozone, little change in March and May 2020 free-tropospheric ozone, and decreases in MAM surface layer CO.

This analysis is a worthwhile and appreciated effort to quantify the effects of COVID emissions changes on the trace gases ozone and CO – there are few published studies that use in-situ profile measurements during the spring 2020 period that aim to accomplish this.

My main concern with this manuscript is that it is difficult to tell how robust, particularly for the surface layer ozone increases, the 2020 anomalies truly are. There are drastically different periods used to calculate the 2020 anomalies for ozone (1994-2019) and CO (2016-2019). The reasoning behind using the 4-year baseline for CO is the decreasing trend since measurements began in 2001. This makes sense. However, there is also a clear increasing trend in surface layer ozone since 1994 (Figure 3b). One assumes that this is at least partly the result of decreasing titration of ozone by NO from long-term NOx emissions reductions. If using a 2016-2019 baseline period to calculate surface layer ozone anomalies, the March 2020 positive anomalies may disappear entirely, and the May anomalies will likely be reduced substantially. 1994-2002 appear to have a strong influence on the 2020 positive ozone anomalies. The results are also shown for months with approximately half of the typical number of profiles. There are no statistics presented on confidence intervals/p-values to confirm the significance of the results and whether they fall outside of expected recent interannual variability.

The results presented here underscore the difficulty of quantifying COVID-related air pollution changes from a single location. The vast majority of published studies of surface and satellite data use dozens to hundreds of locations to bolster their results. It is why Steinbrecht et al. (2020) required dozens of ozonesonde stations to show a convincing decrease in NH free-tropospheric ozone from sonde measurements.

*>We have addressed the reviewer's comments in two ways. First, we have changed our reference period for ozone to 2016-2019 as the reviewer suggested. In addition we have added confidence limits to all of the individual monthly bar plots that we discuss calculated on the basis of Student's t-test. This is such that we can now account for the differing number of profiles in each month. The ozone anomaly in the surface layer in May lies outside the expected recent interannual variability even when the smaller sampling is considered.*

Minor Comment: Were NOx measurements also available on these flights (Berkes et al., 2018; https://amt.copernicus.org/articles/11/3737/2018/amt-11-3737-2018.pdf)? Even if only recent years are available, you could calculate Ox = ozone + NO2. If Ox is about the same in 2020 as past years, and NO2 or NOx is lower, that would further support the argument of reduced NO titration leading to increased ozone in 2020. At the very least, the profile data should be combined with nearby surface NOx data to confirm the NO titration argument, rather than leave it to speculation.

*>Unfortunately, we do not have the NOX data available from IAGOS for these flights as the instrument did not fly during this period. The decrease of NO2 at Frankfurt from surface stations and satellites has already been documented by Barré et al 2021. Barré et al controlled for meteorological factors and the estimates of lockdown-induced NO2 changes for Frankfurt were -24% and -33% based on TROPOMI observations and surface stations respectively. We infer therefore that the positive anomaly in ozone at night is linked to this drop in NO2 at Frankfurt and the consequent reduction in ozone titration. We add reference to this article in the text.*

To summarize, I suggest the following analyses in addition to other topics raised in the line-by-line comments:

- Re-assess the ozone results with the same baseline period as CO of 2016-2019
  *>We have changed our reference period for ozone to 2016-2019 as the reviewer suggested.*
- Produce a more robust statistical analysis indicating the significance of observed ozone and CO anomalies. This is important because only one location is being analyzed, and there is a lot of noise, interannual variability, and underlying long-term ozone trends in the data.

  *>We have added confidence limits and the interannual variability to all of the individual monthly bar plots that we discuss calculated on the basis of Student's t test. This means we can identify which anomalies were significant once sampling has been accounted for.*

- Attempt to incorporate nearby surface NOx/NO2 measurements to confirm the reduction in NO titration of surface layer ozone (or IAGOS NOx if available).

  *>As already mentioned above, we do not have the NOX data available from IAGOS for these flights as the instrument did not fly during this period. The decrease of NO2 at Frankfurt from surface stations and satellites has already been documented by Barré et al 2021. Barré et al controlled for meteorological factors and the estimates of lockdown-induced NO2 changes for Frankfurt were -24% and -33% based on TROPOMI observations and surface stations respectively. We infer therefore that the positive anomaly in ozone at night is linked to this drop in NO2 at Frankfurt and the consequent reduction in ozone titration.*

- Integrate boundary layer CO to account for changes in boundary layer height that potentially reduce the surface layer CO mixing ratios in 2020.

>*We integrated the CO over the boundary layer as the reviewer suggested. This leads to a negative anomaly of 30ppbv in February and 12ppbv in March and May which can be ascribed to the drop in emissions. We have updated the text accordingly.*

[Figure]

**Recommendation:**

This paper could be considered for publication in AMT if the authors present more compelling evidence that the IAGOS ozone and CO data collected in MAM 2020 were directly influenced by COVID-related emissions changes, and not simply a result of interannual variability, long-term trends in ozone (surface layer increases) and CO (decreases), and meteorological factors (e.g. boundary layer heights). I recommend Major Revisions that include an assessment of the statistical significance of the results.

**Specific/Technical and Line-by-Line Comments:**

Line 26: Cite Liu, F. et al. (2020) paper for China TROPOMI NO2 decreases: https://advances.sciencemag.org/content/6/28/eabc2992

>*Done*

Line 27: Cite Duncan et al. (2016) paper, which describes the relationship between economic downturn and NOx emissions/OMI NO2 satellite measurements: https://agupubs.onlinelibrary.wiley.com/doi/full/10.1002/2015JD024121

 >*Done*

Line 26 and/or Line 197: Cite Goldberg et al. (2020) paper, which controls for meteorological variability when examining COVID-related TROPOMI NO2 decreases, https://agupubs.onlinelibrary.wiley.com/doi/full/10.1029/2020GL089269

>*Done on line 33*

Line 79: Change "balloon and sonde measurements" to "balloon-borne ozonesonde measurements"

>*Done*

Line 110: Stylistic comment, suggest to remove "life"

*>Done*

Line 120: Small typographical error "1° horizontal"

*>Done*

Line 125: Please define GFAS acronym

*>Done*

Line 127: "anthropogenic *sources*"

*>Done*

Line 144: How many profiles in total are averaged into the MAM 1994-2019 ozone climatology? Similarly, if there were no profiles in April 2020, how many of the 84 profiles were from March and May 2020? It might be more proper to indicate March/May rather than MAM in the text and figures. (Also see General Comment about the chosen baseline period for ozone).

*>The MAM profile plots now clearly state that this is a March+May average for both the 2020 and the reference period. We have added this to the figure caption. We have also added the number of profiles (220) to the caption.*

Line 181: change reservoir to emissions

*>Done*

Line 189: "in the amount of NO as evidenced by the TROPOMI satellite measurements of NO2"

*>Done*

Line 198: I don't understand what is meant by "but that the photochemical effects from NOx were dominant." Is this just referring to reduced titration of ozone from NO? Please clarify.

*>We have clarified this in the text. The authors found changes in NOX that could not be explained by meteorology alone, and that were a result of the emissions reductions. All found that there were important and differing impacts of meteorology, but that there were changes in NOx that were unattributed to the meteorological conditions and linked to falling emissions during the lockdowns.*

Figures 4 and 5: Is it correct that there are 7 nighttime profiles and 13 daytime profiles in May 2020? How robust is the result of a 41% increase in nighttime surface layer ozone from 7 profiles?

*>It is correct that the number of nighttime profiles is 7 and the number of daytime profiles is 13. We now see a 30% increase in the ozone with respect to the shorter reference period. We have added 95% confidence limits to figures 4 and 5 to show how robust the increases were despite the smaller number of profiles.*

Line 232: change "seen by balloons and sondes" to "observed by ozonesondes"

*>done*

Line 235-236: change "balloon and sonde" to "ozonesonde"

>done

Line 236: Suggest to add: "For example, there was no notable decrease in free tropospheric ozone in the sparsely-sampled Southern Hemisphere."
*>done*

Line 245: Change "inflexion" to "inflection"

*>Done*

Figure 8: (Similar to comment for Line 144) Please indicate how many CO profiles are available for MAM 2016-2019

*>We have added the number of profiles to the figure caption (300 profiles).*

Line 265: Change "biased low" to "anomalously low"

*>Done*

Line 270: Now I see that there are CO profiles for April 2020, so it would be helpful to indicate if the ozone instrument was inoperable in April 2020 (or whatever the cause is).

*>The ozone instrument was not working during April 2020. We have added a line in the text on this.*

Paragraph near Line 280: Is all of this discussion necessary? Isn't the boundary layer height simply calculated at the grid point closest to the Frankfurt airport, where all of the surface layer data are collected? The ECMWF output is a fairly coarse 1° resolution, so I would assume this is the case.  Please correct me if I am wrong.

*>Yes it is approximately the case. The ECMWF data is interpolated to the position of the aircraft. It will have travelled about 20-50km from the airport over the first 2000m of the atmosphere (Petetin et al 2018).  We used a bilinear interpolation in space using a distance weighting from the 4 nearest grid cells to the aircraft position, and a linear interpolation in time. Unless the aircraft is in the centre of the grid cell then the 4 surrounding cells are used. We have added this in the text.*

Line 284: Why not be consistent in definitions of day and night as for ozone (10:00-18:59 UTC and 00:00-09:00/19:00-23:59 UTC)?

>*The referee is right that we did not use the same definition of daytime and nighttime for ozone as for CO, but the objectives of the investigation are not the same. For the ozone, we wanted to account for any bias introduced by an uneven sampling over the diurnal cycle of ozone. For CO we wanted to check the influence of the depth of the boundary layer.*

*We based our definition of the maximum and minimum phases of the diurnal cycle on the diurnal cycle of ozone over Frankfurt as shown by Petetin et al 2016a. The diurnal cycle of ozone depends on UV chemistry and dynamics. This was perhaps not well explained so we have added clarification in the text about this.*

*For CO we verified the impact of the depth of the boundary layer on the anomalies of CO as Peuch et al. 2020 found that the boundary layer was anomalously high during the period of interest. The night and daytime heights of the boundary layer are based on dynamics but not on UV chemistry.*

Lines 289-290: Given the relatively long lifetime of CO, a simple check on whether decreased CO concentrations near the surface are a result of dilution in a deeper boundary layer would be to integrate the boundary layer CO content. This will simplify discussion and may lead to more convincing results.

>*We integrated the CO over the boundary layer as the reviewer suggested. This leads to a negative anomaly of 30ppbv in February and 12ppbv in March and May. We are now more confident that this the result of a drop in emissions. We have updated the text accordingly.*

[Figure]

Line 302: Why are you also excluding fire sources of CO? What does that have to do with lockdown decreases in emissions?
>*We can say a bit more about the fire sources of the CO. It should be noted that SOFT-IO does not calculate a background value for CO. It is adapted to analysing the origin of plumes that are well defined against the background. This is not our case here. We have quantified the absolute contribution from the biomass burning as requested. However, because the anthropogenic emissions are not updated for the COVID period, we cannot give the relative contributions of biomass burning and anthropogenic emissions. Since the biomass burning contribution decreased in 2020, we expect a higher contribution from anthropogenic sources. We have added some discussion about this in the text related to line 302 and line 336 in the comment below.*

Line 304: Please indicate in the text that the trajectories terminate at Frankfurt in the surface layer (>950 hPa).

>We have clarified in the text that the trajectories terminate at the aircraft position within the surface layer.

Figures 12 and 13: Please number the regions on the map and legends so it is easier to identify the source regions. It's a bit difficult to distinguish some of the colors.

*>We have added texture to some of the areas to help distinguish the regions.*

Lines 336-337: Including information from MACC-City fire source CO should help confirm this hypothesis.

*>Yes, in the reference period, the CO from fire sources contributed to 20% of the total, so anthropogenic sources were the primary contribution to CO over Europe. In 2020, absolute amounts from biomass burning have decreased, suggesting a greater contribution from anthropogenic sources. We have added some comments in the text regarding this.*

Line 363: The sign in front of 2 ppbv and 1% should be negative, and actually -1.8 ppbv to be consistent with the previous text (Line 315). > *DONE*

Powered by TCPDF (www.tcpdf.org)

---

## Author Comment (AC2)

**Response to reviewer 2**

The Effects of the COVID-19 Lockdowns on the Composition of the
Troposphere as Seen by IAGOS
Clark *et al*
MS No.: acp-2021-479

**Summary**

The authors use ozone, CO and meteorological measurements from the IAGOS
dataset for Frankfurt airport to study the impact of COVID lockdowns on air quality.
They compare measurements during spring 2020 (European COVID lockdown) to
the previous 27 years of data (20 years for CO). They find surface level increases of
ozone during the lockdown period driven by increases in nighttime ozone. The
anomaly in ozone turns to a slight negative at higher altitudes.
The authors also find a reduction on CO at the surface and little to no reduction at
higher altitudes which they attribute to incoming emissions from outside Europe. This
hypothesis is strengthened by IASI CO retrievals of the same period.

**General Comments**

This is a decent study that fits well with other COVID air quality related publications.
It is useful to have the view of the free troposphere as most publications focus on
surface level impacts. The authors give a good overview of the current literature and
highlight what extra information this study can give. I feel the following comments
need to be addressed before publication.

As the ozone data for April 2020 is missing, is a comparison to the MAM climatology
valid? Do you remove April from the previous years to take the climatology and if not,
does it have an impact on the result?

>*We did not have ozone data for the month of April and April has been removed from the
climatology. The text and figure caption have been updated to make this clearer.*

I find the difference in time series and calculated climatology between ozone and CO
leads to an incoherent picture. I would recommend using the same baseline years
for both ozone and CO.

>*As suggested by the reviewer, we have re-done the analysis for ozone using the same 2016-
2019 baseline as for CO. This has only slightly changed the results, such that there is a
slightly stronger decrease in ozone seen in the free troposphere.*

A measure of statistical significance is needed on the presented results. Without this,
the arguments for why we are seeing the anomalies aren't as strong as they could
be. I am unsure if using the standard deviation as the definition for interannual
variability in this case is the best way to present the results. Would the range or a
percentile (e.g. 95th) be a more suitable parameter to use? If a month is one
standard deviation from the mean there is still ~30% chance it would be completely
expected. Showing if the lockdown period fell outside the 95th percentile for example
would really highlight if it was an unusual year.

*>We have added confidence limits and the interannual variability to all of the individual monthly bar plots that we discuss calculated on the basis of Student's t test. It is now more apparent how significant the anomalies are, since the number profiles within each month is taken into account. The interannual variability over the 2016-2019 period is the standard deviation of the monthly means. Since for the reference period there are only 4 monthly means, the standard deviation seems to be a better choice than percentiles for measuring variability of a sample of size 4.*

Some of the discussion points are mixed in with the results (e.g. the discussion about previous ozone events during heatwaves). This is ok to put the results into context but it leaves the final section of the paper more like a summary rather than a conclusion. I recommend moving some of these sections into the conclusion.

*>As the reviewer points out, some of the discussion in the results section does help to put the results in context and guide the reader. As suggested we have moved some of the discussion on the 2003 heatwave to the conclusions section.*

Section 2.2.2 discusses the drop in CO in the free troposphere and trying to relate that to the lockdowns but as far as I can see from figure 8 they were mostly well within the expected interannual variability. This doesn't appear to be mentioned and the results are presented in a way that suggests that the changes are abnormal. Clearly stating the anomalies are within the expected range is needed here.

*>We have clarified in the text that there was no significant anomaly for CO in the free troposphere. This is made much clearer by the addition of the confidence limits on figure 14.*

**Specific & Technical Comments**

There are a number of occasions where there are double negatives (e.g. -XX% drop). Either say XX% drop or -XX% change (or something similar).

*>We have fixed these*

Throughout: There are a number inconsistencies with spaces between numbers and units which need to be fixed.

*>We have fixed these.*

Line 23: The Le Quere paper only focuses on CO2, I would either add examples of AQ papers by for the atmospheric composition statement, or rewrite this sentence. OK,

*>As suggested, we have added some example of the AQ articles.*

Line 79 & 232: should be 'balloon-borne ozonesondes'

*>Done*

Line 80: I think this should be 'reduction in surface emissions' not 'pollution'

*>Done*

Figure 1 & 8: I suggest changing the y axis to hPa and mark where your free troposphere definitions begin/end. This way the reader can more easily see which bit of the profile you are discussing.

*>We have added horizontal lines to the profiles to highlight the sections discussed.*

Lines 156 & 157: Specify air traffic

*>Done*

Lines 156 – 159: is the 50% reduction in (air) traffic related to the the introduction of restriction measures (22^nd March) or were there measures in place before this date that was reducing air traffic?

*>In the text we have added the following: This reduction was driven by a fall in passenger numbers as lockdown measures spread around the world.*

Line 212 & figure 5: 00:00-09:00/19:00-23:59 is confusing, I would either replace the slash with an ampersand or say '7 pm to 9 am'

*>Done*

Figure 4: What do the numbers on the secondary y-axis denote?  *>The secondary axis denotes the counts for the period 2016-2019. We have added this explanation to the figure caption.*

Line 245: This should be 'inflection' not 'inflexion'

*>done*

Line 262: I would say 'As with ozone...' not 'As for ozone...'
*>done*

Line 291: airports *than* might be though

*>done*

Line 302/303: What do you mean you're ignoring fire or anthropogenic sources of CO? That doesn't leave much. Or do you mean you are not distinguishing between them and only looking at source regions? This needs to be clearer.

>*We are focussing on the geographical source regions of the airmasses in 2020 compared with the reference period but we can say a bit more about the fire sources of the CO. It should be noted that SOFT-IO does not calculate a background value for CO. It is adapted to analysing the origin of plumes that are well defined against the background. This is not our case here. We have quantified the absolute contribution from the biomass burning as requested by reviewer 1. However, because the anthropogenic emissions are not updated for the COVID period, we cannot give the relative contributions of biomass burning and anthropogenic emissions. Since the biomass burning contribution decreased in 2020, we expect a higher contribution from anthropogenic sources. We have added some discussion about this in the text related to line 302 and line 336 in the comment below.*

Line 305: there is reference to fig 13 before saying you're talking about figure 13.

>*We have corrected this.*

---

## Author Response (AR2)

Response of Review of Revision 1 of acp-2021-479 "The Effects of the COVID-19 Lockdowns on the Composition of the Troposphere as Seen by IAGOS at Frankfurt"

Hannah Clark, Yasmine Bennouna, Maria Tsivlidou, Pawel Wolff, Bastien Sauvage, Brice Barret, Eric Le Flochmoën, Romain Blot, Damien Boulanger, Jean-Marc Cousin, Philippe Nédélec, Andreas Petzold, and Valérie Thouret

The authors would like to thank both reviewers for their thorough and constructive comments on this article. Here is our point-by-point response to the corrections on the review of revision1.

Specific/Technical and Line-by-Line Comments:

Line 26: Formatting error on the Duncan et al. (2016) reference.

This was an error in the latex citation which we have corrected.

Figure 1 Legend: Change "Mars" to "Mar"

This has been done

Line 284: Text error remains where there is a strikethrough of "for", replaced by "with"

Another latex issue which we have corrected.

Line 288: Here and Figure 10 caption, you're showing 2001-2020 not just 2016-2020, correct?

Yes, that is correct. We are showing 2001-2020 and have updated the caption accordingly.

Line 307: For each year between 2016 and 2020?

Yes, we have corrected this.

Line 326: "This makes"

We have added the missing 's'.

Line 361: Are these 3 and 2 ppbv values the mean biomass burning contributions to the anomaly? I'm not sure how to interpret the "3 ppbv or 20%" values indicated here. Do you mean 2%?

Yes, these are the contributions to the anomaly. We have re-written line 361 to make this clearer. The '2%' referred to by the reviewer, is 2ppbv as stated in the text. We have quoted an absolute amount for 2020 because the uncertainites in the anthropogenic emissions in 2020 mean that we cannot calculate a reliable relative amount.

Figure 14 Caption: I think this has the same issue as noted for Figure 10, which also shows 2001-2020, not 2016-2020. Please check.

Yes, the figure caption has been updated.

Line 378: To be precise, neither day nor night May SL ozone anomalies are statistically significantly different from the interannual variability in Figure 4. Suggest removing the word "significant", or moving it to the sentence above, as you've shown in Figure 2 that the 32% increase for all of May is significantly different from interannual variability.

As the reviewer suggested, we have moved the word "significant" to the sentence above (now line 381) and reworded slightly the sentence.

We also detected a mis-numbering of the sections which we have corrected in this revised version.